

# A multiple spatial scales water use simulation for capturing its spatial heterogeneity through cellular automata model

Jiayu Zhang[a], Dedi Liu*[,a,b,c], Jiaoyang Wang[a], Feng Yue[a], Hanxu Liang[a], Zhengbo Peng[a], Wei Guan[a]

[a] State Key Laboratory of Water Resources Engineering and Management, Wuhan University, Wuhan, China

[b] Hubei Provincial Key Lab of Water System Science for Sponge City Construction, Wuhan University, Wuhan, China

[c] Department of Earth Science, University of the Western Cape, Robert Sobukwe Road, Bellville 7535, Republic of South Africa

* Correspondence to: Dedi Liu: dediliu@whu.edu.cn

**Abstract:** Reliable water use simulation is essential for sustainable water resource planning, especially under intensifying pressures from climate change, population growth, and socio-economic transitions. While previous studies have extensively explored water availability as supply side modeling across multiple spatial scales for its spatial heterogeneity, the water demand side remains relatively underdeveloped—often constrained by fixed spatial scales and coarse statistical data that assume spatial homogeneity. This mismatch between supply side and demand side limits the ability of existing models to accurately represent spatial heterogeneity in water use and brings uncertainty into water resource allocation strategies. To address this mismatch, we propose a novel multi-scale water use simulation framework by integrating cellular automata (CA) model with Generalized Likelihood Uncertainty Estimation (GLUE). The CA model captures the spatial heterogeneity of water use through the grid-based update rules. Two update rules are adopted—probability rule (i.e., capturing stochastic transitions via distribution fitting) and linear rule (i.e., modeling neighborhood-weighted evolution). To evaluate the impacts of spatial scale on water use heterogeneity, simulations are conducted at three spatial scales: 1 km, appropriate scale, and prefecture scale across 341 prefectures in China. Results show that both the update rule and spatial scale significantly affect spatial heterogeneity and uncertainty of water use. The probability rule can capture the broader variability but results in higher Root Mean Squared Error (*RMSE*) and Relative Error (*RE*) while the linear rule brings more stable performance with lower errors. While the 1 km scale increases uncertainty due to sensitivity to local fluctuations, and the prefecture scale suppresses spatial details, the appropriate scale offers the best trade-off between stability and spatial heterogeneity. The uncertainty quantified by GLUE, expresses as confidence intervals, varies across prefectures and spatial scales. Overall, the proposed framework offers a flexible tool for multi-scale water use simulation and highlights the critical role of spatial heterogeneity, thereby supporting adaptive water resource planning and management.

**Key words:** water use; spatial scale; cellular automata; multi-scale simulation

# 1 Introduction

Water scarcity has become one of the most pressing global challenges, exacerbated by climate change, population growth, and unsustainable water use practices (Avargani et al. 2022, Huang et al. 2021, Kaewmai et al. 2019, Rosa et al. 2020). Nearly 2.3 billion people is living in regions facing water scarcity (Brunner et al. 2019, Dolan et al. 2021, Mekonnen and Hoekstra 2016). In this context, accurate and timely assessments of water scarcity are essential for effective water management, resource allocation, and policy-making (Avargani et al. 2022, Cao et al. 2018). A scientifically rigorous water scarcity assessment requires a comprehensive understanding of both available water resources and water use (Brunner et al. 2019, Ji et al. 2025, Sun et al. 2022). However, these two components are often represented at incompatible spatial scales, resulting in mismatches that complicate accurate evaluation and integrated water resources planning (Almino and Rufino 2021, Kang et al. 2017).

In the past few decades, significant advancements in hydrological modeling, satellite remote sensing, and reanalysis datasets have enabled researchers to simulate surface and ground available water resources across various temporal and spatial scales—from daily basin-scale runoff forecasts to long-term continental water balance projections (Horta et al. 2024, Su et al. 2024, Yang et al. 2021, Zhang and Long 2021). These advancements have laid the foundation for widely used tools such as SWAT, VIC, H08, and PCR-GLOBWB, which help water managers better understand available water resources under different spatial-temporal scales (Hersbach et al. 2020, Kovacevic et al. 2020, Noori and Kalin 2016, Sunkara and Singh 2022). Though there has been water use simulation progress, it still faces significant challenges and has yet to achieve the same spatial level of sophistication as the simulation of available water resources. The primary limitations of water use simulation lies in the available spatial scales of water use data (Hou et al. 2024, Zhang et al. 2023). Much of the existing researches rely on a coarse, aggregated dataset such as national statistics, sectoral usage reports,

or administrative boundaries (e.g., counties, provinces) (Carvalho et al. 2021, Wu et al. 2022, Zhang et al.
2023). These datasets inherently assume uniform water use within each administrative unit, and often overlook
the spatial heterogeneity of water use. This oversimplification limits the ability of water use simulations to
capture spatial variation, particularly in regions with pronounced heterogeneity (Brunner et al. 2019, Su et al.

2024).

Simulating water use at a grid scale provides a promising solution to address the spatial scale mismatch,

allowing for a more accurate representation of water use dynamics and capturing regional variations (Su et al.
2024, Wu and Lu 2021). A grid-based approach allows for a more accurate representation of water use
dynamics by capturing fine-scale spatial heterogeneity, which administrative survey data often overlook.
Zhang et al. (2023) integrated the Iterative Input Selection algorithm with Convolutional Neural Networks to
simulate annual irrigation water use, producing high-resolution grid maps with a spatial resolution of 1 km for
mainland China. Hou et al. (2024) developed a monthly dataset on industrial water withdrawals, incorporating
spatial resolutions of 0.1° and 0.25° by utilizing enterprise data, product yields, and water use statistics. These
studies have demonstrated that the downscaled water use data from administrative level to grid scale can catch
a more detailed and region-specific representation of water use patterns, thereby improving their applicability
in water resource management and water scarcity assessment. However, these studies still rely on a fixed
spatial grid scale, limiting their ability to capture the full spatial heterogeneity of water use. The effects of
spatial resolution on the representation of water use heterogeneity remain insufficiently explored. In particular,
finer scales (such as 1 km grids) can capture localized variations in water use, coarser scales (such as regional
or administrative boundaries) tend to smooth over spatial differences, potentially obscuring critical patterns
(Luo et al. 2020, Sun et al. 2022). Therefore, simulating water use across multiple spatial scales is essential
for capturing the full spectrum of spatial heterogeneity. By incorporating the results at different spatial scales,
this simulation can account for the fine-scale dynamics of water use in urban, agricultural, and industrial areas
more effectively, offering a comprehensive understanding of water use patterns and improving the accuracy
of water scarcity assessments (Su et al. 2024, Sunkara and Singh 2022).

Since the spatial scale of water use simulation depends on the spatial scale of the input data (Horta et

al. 2024, Sharifi et al. 2021, Zhang et al. 2023), achieving multi-scale water use simulation requires a model
that can flexibly handle different input spatial scales. Such a model should be adaptable to varying spatial
resolutions and ensure accurate simulation of water use, whether the focus is on fine-scale urban areas,
intermediate regional levels, or broader national assessments. The cellular automata (CA) model, with its grid-
based structure, offers a suitable framework for spatially explicit modeling across multiple scales by adjusting
both the spatial resolution of input data and the design of updating rules (Al-Shaar et al. 2022, Sapino et al.
2023, Tariq et al. 2023, Wang et al. 2020). The CA model can thus be employed to support multi-scale water
use simulation. And as uncertainties are inherently associated with the selection of spatial scales and update
rules (Yin et al. 2020, Zhang and Long 2021), it is critical to quantify and address these uncertainties to ensure
the reliability of simulation outcomes.

The aim of this research is to develop a multi-scale water use simulation framework that specifically

addresses the impact of spatial scale on the spatial heterogeneity of water use. The framework is mainly
composed of the CA model for simulating water use dynamics across various spatial scales and an uncertainty
analysis technique for quantifying the uncertainties of the simulation. The proposed framework will not only
facilitate a deeper understanding of the influences of spatial scale on water use heterogeneity in diverse regions,
but also improves the accuracy of water scarcity assessments and supports more effective resource
management. Ultimately, the proposed framework contributes to advancing sustainable water governance in
areas facing pronounced water stress.

## 2. Methodology

To develop a multi-scale water use simulation model, the dynamic spatial scale simulation capabilities
of the CA model should be firstly leveraged. The particularly advantage of CA model is modeling complex
global dynamics through simple local interactions and transition rules (Al-Shaar et al. 2022, Liu et al. 2021,
Tariq et al. 2023). Moreover, the grid-based structure of the CA model allows it to flexibly accommodate
various spatial resolutions, making it well-suited for modeling water use at multiple spatial scales (Sapino et
al. 2023, Wang et al. 2020). In this framework, water use grid maps at different spatial scales are first prepared
as inputs to the CA model. Each grid is treated as an individual cell in the CA model, with the water use
amount at each cell representing its state. Based on the neighborhood configuration and the initial baseline
year water use data, water use values for future years are simulated using predefined update rules for each cell
at multiple spatial scales. The procedure of our proposed CA model for multi-scale water use simulation is
depicted in Figure 1.

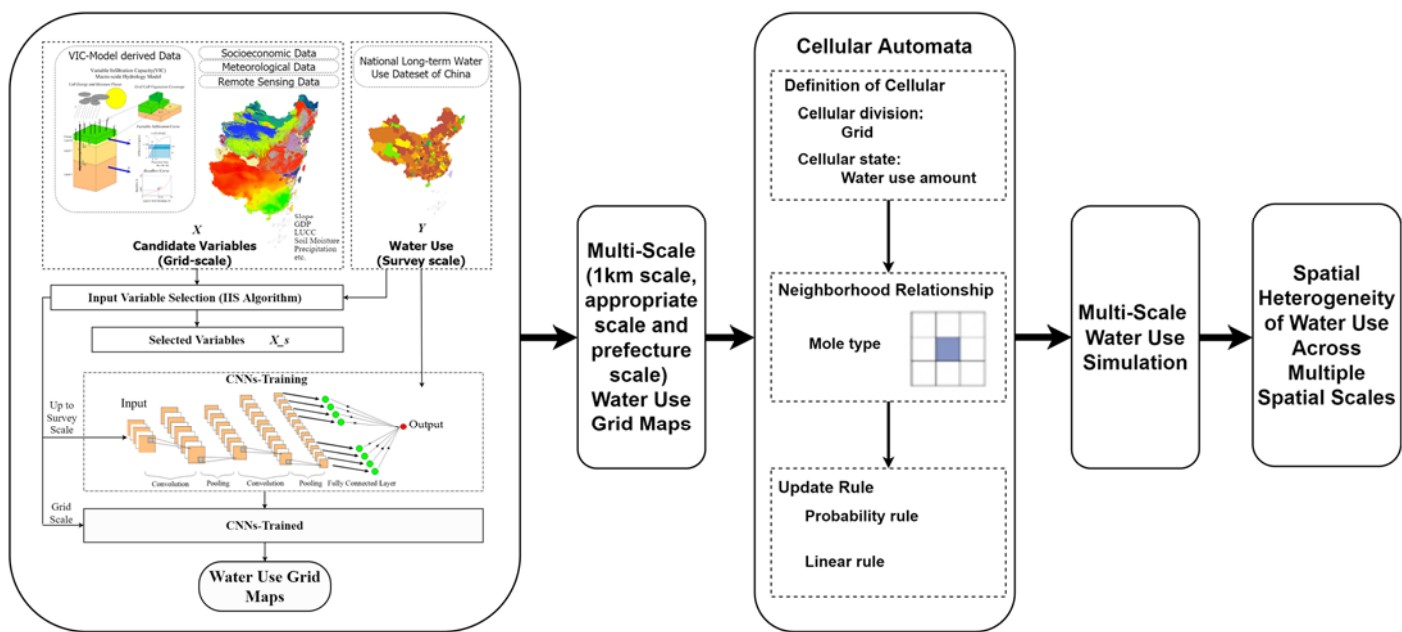


Figure 1. Procedure of the multi-scale water use simulation

To capture the spatial heterogeneity of water use dynamics at different scales, the Coefficient of Variation
(CV) (Abdi 2010) and Moran's I spatial autocorrelation index (Tiefelsdorf et al. 1995) are employed here. The
CV measures the relative variability of water use across regions, with higher values indicating greater spatial
heterogeneity. Moran's I assesses the degree of spatial autocorrelation, identifying whether similar values are
spatially clustered (positive values) or dispersed (negative values). These two indices are applied to evaluate
the spatial heterogeneity of water use grid maps at different spatial scales (i.e., 1 km, appropriate scale, and
prefecture scale), and figure out the influences of the spatial variations on water use dynamics and uncertainty
in the simulation results.
To quantify the uncertainties of water use simulation for providing a more robust foundation for water
scarcity assessment and policy-making (He et al. 2018, Knox et al. 2018, Sharifi et al. 2021), the Generalized
Likelihood Uncertainty Estimation (GLUE) method is adopted here. GLUE provides a probabilistic
framework for model evaluation by exploring a wide range of parameter sets and assigning likelihoods to each
based on their performance. This approach effectively addresses model equifinality—where multiple
parameter combinations yield similar outputs—and is particularly well-suited for complex, non-linear models
such as cellular automata (Liu et al. 2015, Yin et al. 2020).

## 136 2.1 Water Use Grid Maps generating

The spatial scale of water use simulation is determined by the spatial scale of the input data, so water use
grid maps at different spatial scales were prepared as input to the simulation model. Here, water use refers to
the total water consumption across three major sectors: irrigation, industrial, and urban domestic water use.
The water use data considered in this study account for both groundwater and surface water sources (e.g.,
rivers, lakes, and reservoirs). These data were drawn from Zhou et al. (2020), which compiled water use data
across 341 Chinese prefectures. The dataset includes water consumption data for irrigation, industrial, and
domestic uses, incorporating both groundwater and surface water sources. The water use data were sourced

from two major nationally coordinated surveys: the First and Second National Water Resources Assessment

Programs (1965–2000) and the Water Resources Bulletins published by 31 provincial governments (2001–

2013). Both surveys were led by the Ministry of Water Resources of China, and followed consistent

methodologies in terms of definitions, survey units, sector classifications, field measurements, and quality

assurance. To obtain the water use grid maps, several steps should be done to convert the water use data at

administrative survey scale into spatially explicit grids of varying resolutions.

The grid maps of irrigation, domestic, and industrial water use are generated from the prefecture-level

statistical survey data and water use sector-specific predictor variables. For each sector, the most relevant input

variables are identified through an iterative input variables selection algorithm (Zhang et al., 2023; Zhang et

al. 2025). Specifically, irrigation water use was modeled by the potential evapotranspiration, normalized

difference vegetation index ($NDVI$), rainfall and soil moisture; domestic water use was modeled by population,

rainfall, temperature and night-light; industrial water use was modeled by $GDP$, night-light, population and

rainfall. And then these sectoral gird maps were aggregated to form total water use grid maps for modeling.

This aggregation is done for two reasons: the first one is that the temporal trend of the total water use has

become stable during the study period (1998–2013) and future. The average annual growth rate is only about

0.87% (from 505.53 billion $m^3$ in 1998 to 575.44 billion $m^3$ in 2013) due to the policy interventions,

technological improvements, and industrial structure changes. Since the primary objective of our study is to

examine the influence of spatial scale on the spatial heterogeneity of water use, a temporally stable indicator

helps minimize the confounding effects of sector-specific temporal fluctuations; the second reason is that the

total water use can figure out the scale effects across regions instead of the sector-level temporal variability

while the sectoral differences are implicitly in the inputs before the aggregation.

Earlier studies often applied a fixed spatial resolution in different regions, which could not account for

differences in land area, natural endowments, and water use structures, and leaded to the discrepancies in information density and potential over- or underestimation of water use. To address this issue, an appropriate spatial scale can be determined by the deep learning-based spatiotemporal scale adaptive selection model (Liu et al., 2021; Zhang et al., 2025). And the model can balance the accuracy of the simulation based on the spatial information density of gridded water use data, and its results vary across prefectures. The spatial scale selection module in the selection model figures out the appropriate spatial scale by maximizing information density while balancing simulation accuracy in terms of the Conditional Entropy, Kullback–Leibler Divergence Loss and Relative Error performance metrics. This selection module enables each prefecture to adopt its own appropriate spatial scale rather than a fix resolution. The detailed values of the appropriate spatial scale (in km) for each prefecture and water use sector (irrigation, industrial, and domestic) are provided in an accompanying Excel file, which can be accessed and downloaded via the data link: 10.6084/m9.figshare.30445157. This file allows users to examine the spatial heterogeneity of the appropriate scale across regions in detail. Finally, total water use grid maps are generated at three spatial resolutions: the small scale (e.g., 1 km), the appropriate spatial scale as determined by the selection module, and the prefecture scale as the statistical survey water use data.

## 2.2 Cellular automata model for water use simulation

The CA model, grounded in complexity theory, is widely used in land use and urban growth modeling. It provides a robust platform for simulating spatial phenomena governed by local interactions and transition rules (Sapino et al. 2023, Tariq et al. 2023). Each cell in a CA model represents a discrete spatial unit that updates its state over time based on predefined rules and the states of its neighboring cells. It's decentralized, bottom-up modeling structure enables the simulation of complex global behaviors emerging from simple local dynamics (Al-Shaar et al. 2022, Wang et al. 2020). The CA model is introduced to simulate the temporal

evolution of water use at the grid scale, complementing the static spatial distribution obtained from the Convolutional Neural Network (CNN) downscaling. While the CNN model effectively reconstructs the spatial pattern of water use for each prefecture based on physical and socioeconomic predictors (Zhang et al., 2023), it does not explicitly account for the spatial dependence and interactions among adjacent grid cells. The CA model addresses this limitation by incorporating spatial adjacency effects and feedback mechanisms, allowing each cell's water use to be influenced by its neighbors. This enables the model to represent the diffusion and clustering behaviors of water use, which are essential for capturing the spatial heterogeneity and dynamic interactions of human water activities.

Both the probability and the linear update rules are designed and tested to capture the dual nature of water use dynamics. The probability rule has been widely applied in significant spatial and temporal variation areas in land use simulation and other fields. It will be designed here for the water use at different scales. Rather than assuming temporal stability, the probability rule explicitly incorporates the variations through calibrating the state transition matrix and probability distributions for each prefecture independently by the own historical water use record. This rule enables the simulation to capture both the structured temporal dependence and the inherent randomness in water use, ensuring adaptability to local conditions. The linear update rule assumes that changes in water use are more deterministic and can be approximated as a linear combination of the cell's own state and those of its neighbors. This rule is more appropriate for long-term, high spatial autocorrelation, and persistent patterns. After implementing and comparing the water use simulation results of the two rules in the CA framework, their results can assess the relative effectiveness of stochastic versus deterministic update mechanisms across different spatial scales. These two rules not only strengthen the robustness of the modeling framework but also provide insights into the dominant processes shaping water use dynamics in different regions.

## 2.2.1 Probability rule in CA

The probability rule in the CA model is designed to represent the stochastic state transitions of water use over time. It abstracts the temporal dynamics of water use at the grid level into a probabilistic transition framework that can be applied consistently across different spatial scales and regions, while remaining adaptable to significant spatial and temporal variations. This adaptability is achieved by calibrating the update rule separately for each prefecture by its own historical water use record for appropriately capturing both long-term trends and localized fluctuations. In this approach, the state of each grid cell (i.e., representing the amount of water use) is divided into $k$ distinct intervals using equal-frequency categorization based on the cell's historical water use record. This categorization ensures that the intervals reflect the variations in water use over time. For each interval, the most suitable statistical distribution is selected using the Akaike Information Criterion (AIC). The selection process enables the model to represent the probabilistic characteristics of water use within each intensity class. And the distribution is chosen from a set of candidate distributions, including normal, lognormal, exponential, gamma, and uniform.

Once the optimal distribution is found for each interval, a state transition matrix is constructed based on observed transitions of grid cells between intervals from one year to the next. The transition matrix captures the likelihood of a grid cell moving from its current water use state to another in the subsequent time step. The model first generates the next state probabilistically through the transition matrix, and then the water use samples are generated from the corresponding probability distribution. These two steps incorporate both the structured temporal dependence and the inherent randomness in future water use patterns.

In the probability rule, the calibrated parameter is the number of state intervals and is denoted as $k$. The value of $k$ directly affects the granularity of the state categorization and the accuracy of the state transition matrix. A larger $k$ increases the resolution of the state representation and captures the finer variations in water

use, but a larger $k$ can also lead to overfitting. And a smaller $k$ oversimplifies the demand pattern. To calibrate
the parameter $k$, the historical and observed water use data is divided into a calibration and a validation sets.
And the performances of the model with different $k$ values is then evaluated by the Root Mean Squared Error
(*RMSE*) and Relative Error (*RE*) metrics. The optimal $k$ can be calibrated by the minimums of the *RMSE* and
*RE* in the validation period, ensuring a balance between model accuracy and generalizability.

## 2.2.2 Linear rule in CA

The linear rule in CA updates the water use of each cell according to the weighted average of its current
state and the states of its neighboring cells. The linear rule assumes that the water use at a given grid cell is
influenced by both its own previous state and the water use of adjacent cells. The linear rule is expressed as
the Equation (1).
$$W(t+1) = \alpha W(t) + \beta \sum_{i=1}^{n} \omega_i W_i(t) \tag{1}$$

where   $W(t+1)$ is the predicted water use of the central cell at time $t+1$, $W(t)$ is the current water use of the
central cell at time $t$, $W_i(t+1)$ represents the water use at the $i^{\text{th}}$ neighboring cell ( $i=1,2,\ldots,8$) at time $t$, $\omega_i$ is
the weight that is assigned to the $i^{\text{th}}$ neighboring cell, $\alpha$ and $\beta$ are coefficients that control the relative influence
of the central cell's own water use and the neighboring cells' water use.
To accurately capture the influences of neighboring cells, the weight $\omega_i$ for each neighboring cell is
determined by an inverse distance weighting scheme with an exponential decay and is expressed as the
Equation (2).
$$\omega_i = \frac{1}{d_i^p} \tag{2}$$

where $d_i$ is the Euclidean distance between the central cell and the $i^{\text{th}}$ neighboring cell, $p$ is an adjustable
exponent that controls the rate of decay in the influence of neighboring cells.
To determine the parameters $\alpha$, $\beta$, and $p$, the observed period is also divided into a calibration period and
a validation period. After the model runs with different parameter combinations, and their performances are
assessed in terms of *RMSE* and *RE*, the optimal parameter set will be picked out by the minimum of the
performance metrics in the validation period.

## 2.3 Generalized Likelihood Uncertainty Estimation for water use simulation

Generalized Likelihood Uncertainty (GLUE) is a probabilistic approach that evaluates a model's
performance by considering a wide range of plausible parameter sets and quantifying the likelihood of each
set in its ability to reproduce observed data (Chen et al. 2011, Sharifi et al. 2021, Taormina et al. 2016). Unlike
traditional deterministic calibration methods, GLUE acknowledges the inherent equifinality in environmental
modeling that also is the possibility of the acceptable results from the multiple parameters set. The GLUE is
incorporated into the CA model to assess the uncertainty in water use simulations.
The GLUE is applied in both the probability rule and the linear rule of the CA model to evaluate the
uncertainty of water use simulations. There are six steps. (1) Parameter Selection: the number of state intervals
$k$ for the probability rule, and the self-influence coefficient $\alpha$, the neighboring influence coefficient $\beta$, and the
distance decay exponent $p$ for the linear rule; (2) Parameter Sampling: a large number of parameter sets are
generated by the uniform sampling within specified ranges for each parameter; (3) Model Simulation: the CA
model is executed by each parameter set to simulate water use for the target period; (4) Likelihood Estimation:
*RMSE* and *RE* values are calculated for each simulation to evaluate how well the simulated results match the
observed data; (5) Behavioral Parameter Identification: behavioral parameter sets can yield acceptable
likelihood values according to the thresholds that is determined by the calibration data; (6) Uncertainty
Quantification: from the range of outputs from the behavioral parameter sets, a prediction interval is
constructed quantify the uncertainty associated with the water use projections.

## 2.4 Spatial Heterogeneity of water use

To comprehensively understand the variability and spatial relationships of water use patterns across
different spatial scales, Coefficient of Variation (CV) (Abdi 2010) and Moran's I (Tiefelsdorf et al. 1995) are
used to analyze the spatial heterogeneity of water use. CV is defined as the ratio of the standard deviation to
the mean of water use values at each spatial scale, offering a normalized measure of dispersion. It is a primary
indicator used to quantify variability in water use across grid cells (Canchola et al. 2017). A higher CV value
indicates greater spatial variation in water use, while a lower CV suggests more uniform water use patterns
across the region. Once the CV for each grid cell is calculated, the extent of water use spatial heterogeneity is
assessed, and areas with high variability are identified as more susceptible to water stress and fluctuations in
demand (Botta-Dukát 2023, Liu et al. 2020).
Moran's I can reflect the spatial autocorrelation by measuring the degree to which one grid cell's water
use is similar to that of neighboring grid cells. Moran's I provides an overall measure of spatial dependence,
where a positive Moran's I indicates a clustering of similar water use values (either high or low) in neighboring
grid cells, while a negative Moran's I suggests a dispersed pattern. A value close to zero indicates a random
spatial pattern with no significant clustering (Gedamu et al. 2024, Shortridge 2007). By calculating Moran's
I across different spatial scales, we are able to detect whether water use patterns exhibit spatial clustering or
they are more randomly distributed. Moran's I can therefore help us understand the regional patterns of water
use and identify areas that require targeted management interventions (Fu et al. 2024, Yamada 2024).
Both the CV and Moran's I provide a robust framework to analyze spatial heterogeneity in water use.
The CV provides a measure of variability, while Moran's I reveals the extent of spatial correlation in water
use. They offer a more comprehensive understanding of the spatial dynamics across different regions.

# 3 Study area and datasets

## 3.1 Study Area

Situated on the northwestern shore of the Pacific Ocean, China boasts vast territory, a large population, and diverse climate conditions (Ji et al. 2025, Sun et al. 2022). Over the years, the volume of water use in China has surged significantly rising from 3305 $km^3$ in 1965 to 5925 $km^3$ in 2024 (Ji et al. 2025). Owing to the country's varied climate conditions and pronounced spatiotemporal heterogeneity of water resources, water scarcity has become a recurrent challenge, exacerbated by the looming threats of climate change and rapid socio-economic development (Hou et al. 2024, Wang et al. 2021). Notably, China is expected to experience greater impacts from climate change than the global average (Kang et al. 2017, Sun et al. 2022, Yan et al. 2019). Therefore, conducting a multi-scale water use simulation study in China is particularly meaningful due to the country's complex water resource challenges, amplified by regional disparities, climate change, and rapid socio-economic development. China comprises 31 provincial-level administrative divisions, including provinces, autonomous regions, and municipalities. As outlined in the study by Zhou et al. (2020), the administrative divisions are categorized at the prefecture level, totaling 341 prefectures (Figure. 2).

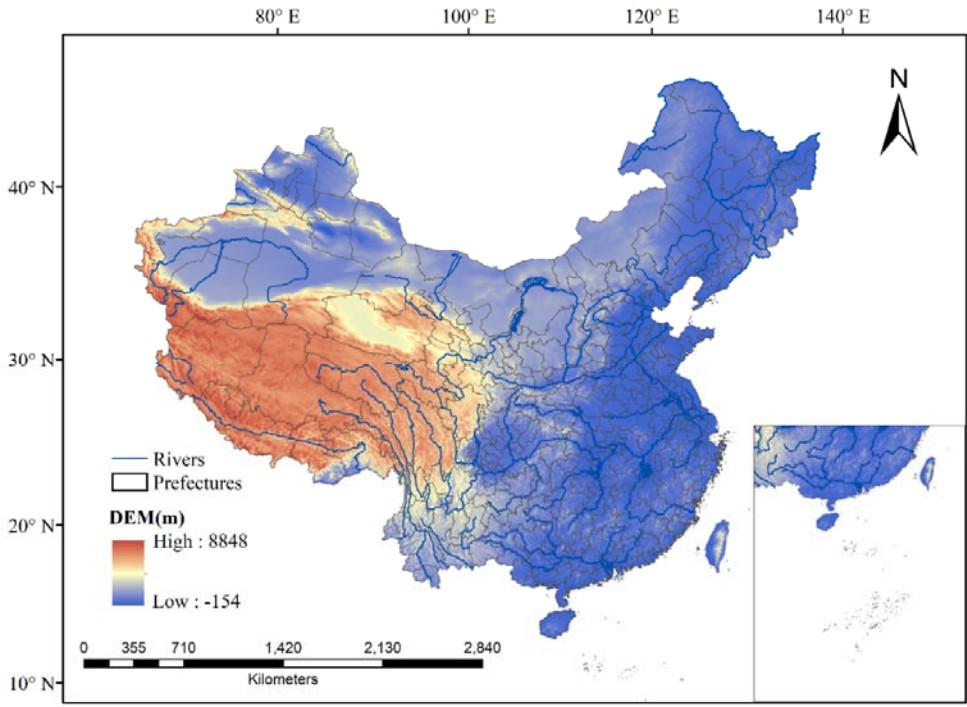


Figure. 2 Prefectures and major rivers in study area

## 3.2 Datasets and Preprocessing

To obtain water use grid maps, the observed datasets related to irrigation water use, domestic water use,

and industrial water use were collected. The observed datasets are composed of the annual water use statistical
survey data at the administrative scale, soil moisture data derived from hydrological models, socio-economic
data like *GDP* (Gross Domestic Product) and population, meteorological data from point observations, and
satellite remote sensing data including the normalized difference vegetation index (*NDVI*) and night light data.

Various spatial interpolation and downscaling methods are employed to transform the datasets into

different spatial scales. To simulate water use at different spatial scales, we transform the spatial scale of the
dataset to 1km, appropriate spatial scale (Zhang, et al., 2025) and prefecture scale. The details of the transform
methods can be found in the reference of Zhang et al. (2023). Nighttime light remote sensing data is also
combined with the land use information to perform upscaling or downscaling at spatial scales (Ye et al.
2021).All the adopted datasets and their corresponding preprocessing methods are listed in Table 1.

Table 1. Datasets and corresponding preprocessing methods

| Variables | Data Sources | Origin Data Format | Origin Resolution | Processing Method |
|---|---|---|---|---|
| Precipitation (P) | CMA* | Point observation | \ | Interpolation (ANUSPLIN) |
| Temperature (T) | CMA* | Point observation | \ | Interpolation (IDW) |
| LUCC | RESDC** | Grid | 30 m × 30 m | Upscale (Resample) |
| NDVI | RESDC** | Grid | 1 km × 1 km | Upscale/downscale (Resample) |
| GDP | RESDC** | Grid | 1 km × 1 km | Upscale/downscale (Resample) |
| Population | RESDC** | Grid | 1 km × 1 km | Upscale/downscale (Resample) |
| DEM | SRTM | Grid | 1 km × 1 km | Upscale/downscale (Resample) |
| Potential Evapotranspiration (PET) | \ | \ | \ | P-M Equation/Interpolation (IDW) |
| Soil moisture (SM) | GDFC*** (He and Sheffield 2020) | NetCDF | 0.25° × 0.25° | Downscale (Machine learning)(Guevara and Vargas 2019) |
| Night light (N-L) | DMSP-OLS | Grid | 2.7 km × 2.7km | Upscale/downscale (Resample) |
| National Long-term Water Use Dataset of China | FigShare-PNAS(Zhou et al. 2020) | Excel | \ | \ |

CMA*: China Meteorological Administration (http://data.cma.cn/); RESDC**: Resource and Environment Science and Data Center (https://www.resdc.cn/).

GDFC***: Global Drought and Flood Catalogue (http://hydrology.princeton.edu/data).

# 4. Results

## 4.1 Water use simulation by CA model

### 4.1.1 Water use simulation from the probability rule CA model

In the CA model with the probability rule, the number of state intervals ($k$) is the only parameter to be calibrated. The dataset from 1998–2009 is used for calibration and 2010–2013 for validation, with *RMSE* and *RE* as performance metrics. The optimal value of $k$ at three spatial scales (1 km, appropriate scale, prefecture scale) for each prefecture is determined by minimizing *RMSE* and *RE* in the validation period. The calibrated $k$ values for each prefecture, along with the corresponding *RMSE* and *RE* during the calibration (1998–2009) and validation (2010–2013) periods at the three spatial scales, are presented in Figure 3.

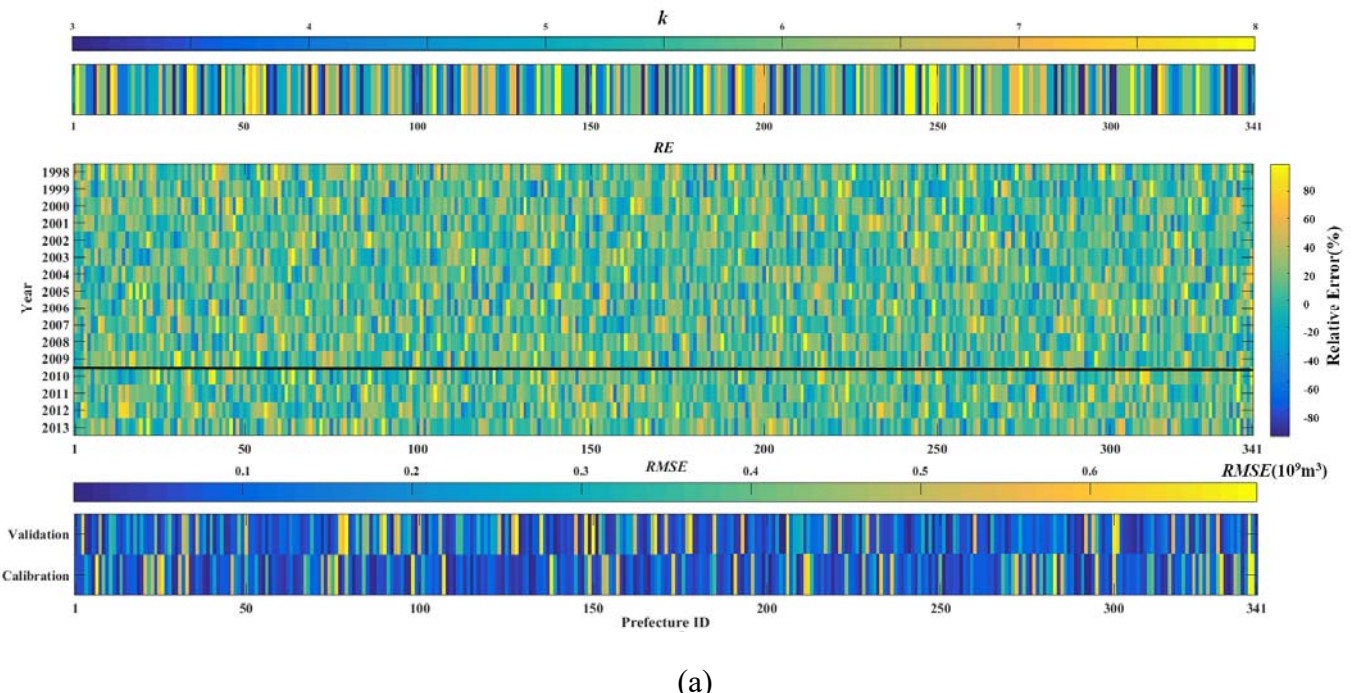

(a)

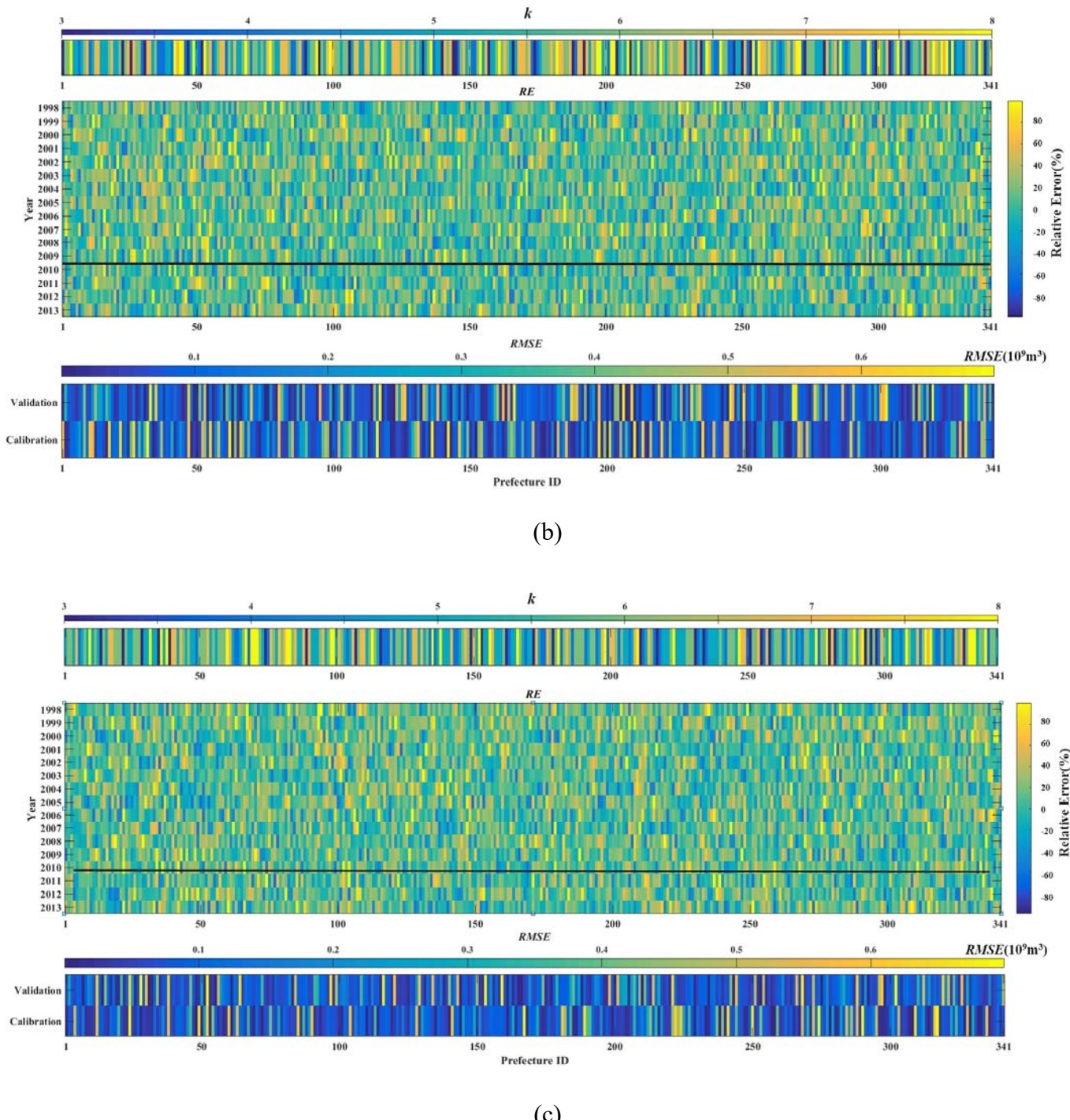

Figure .3 Optimal parameters of the probability rule CA model and the model performances at: (a) 1km

scale; (b) appropriate spatial scale; (c) prefecture scale

According to the results shown in Figure 3, the calibrated parameter $k$ exhibits clear spatial heterogeneity

across prefectures and varies with spatial scales. At the 1 km scale (Figure 3(a)), most prefectures show $k$

values concentrated around 5–6, corresponding to relatively low $RMSE$ and $RE$ values. This suggests that a

moderate number of state intervals can effectively capture local water use variability while avoiding
overfitting. In these areas, the probability distributions and transition probabilities appear to reflect stable
temporal patterns, resulting in more accurate simulations. At the appropriate spatial scale (Figure 3(b)), the
distribution of $k$ becomes more diversified among prefectures. Some regions require larger $k$ values (>=7) to
preserve finer distinctions in water use states, while others perform better with smaller $k$ values (<=4) that
smooth out excessive variability. Different from the results at the 1 km scale, the overall *RMSE* and *RE* values
are slightly higher, indicating that while the appropriate scale balances detail and generalization, it may not
fully capture abrupt local changes in some prefectures. At the prefecture scale (Figure 3(c)), $k$ values are
generally smaller (mostly 3–4), reflecting the reduced spatial detail at coarser resolution. Thus, their accuracies
of the simulation decrease, with higher *RMSE* and *RE* values. When the input variability of small scale is
strongly aggregated at large scales, fewer state intervals oversimplify the temporal transitions, leading to
greater deviations from observed water use patterns.
After determining the optimal $k$ values at each scale, the next step is to characterize the statistical nature
of water use within each state interval. The Akaike Information Criterion (AIC) is taken as performance metric
to select the most suitable probability distribution for each interval in every prefecture. The AIC can balance
the model fitness and the complexity through penalizing excessive parameters, it can reduce the risk of
overfitting. The selected distribution types not only fit the historical data well but also is used to generate the
future scenarios. The complete AIC values and corresponding best-fitting distribution types for each prefecture
are provided in an open-access Excel file, which can be downloaded from: 10.6084/m9.figshare.30445157.
The results of the optimal probability distributions for water use grids at the three different spatial scales (i.e.,
1 km scale, appropriate spatial scale, and prefecture scale) are shown in Figure 4. These distributions,
combined with the calibrated $k$ values, form the basis of the probability rule CA model's ability to reproduce
the spatial and temporal heterogeneity of water use.

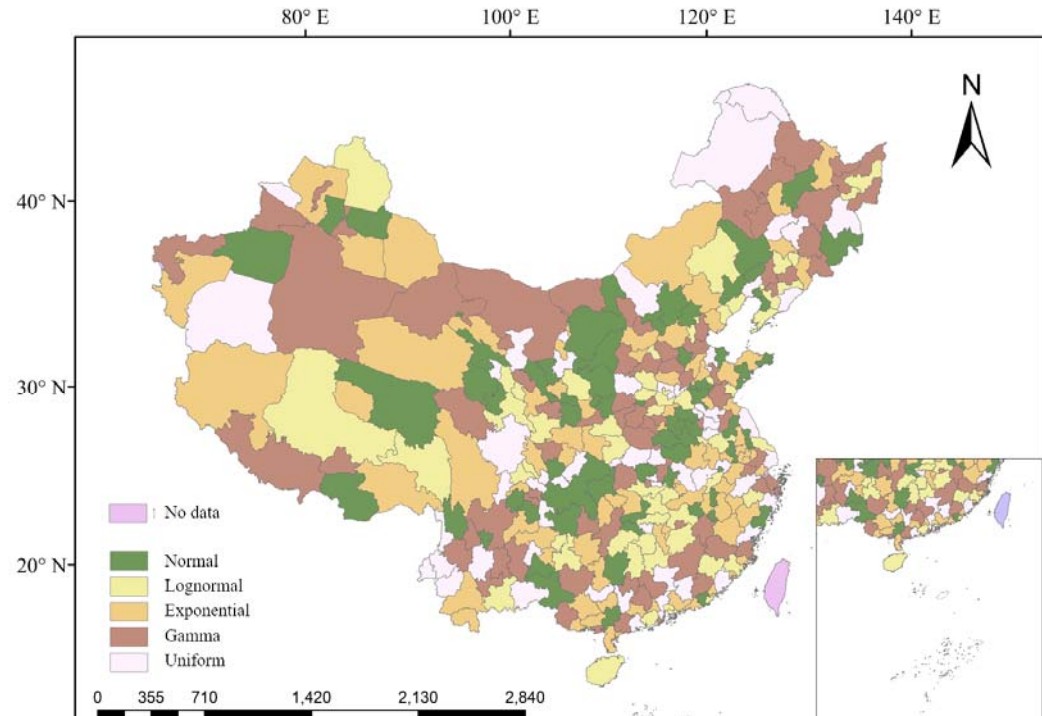


(a)

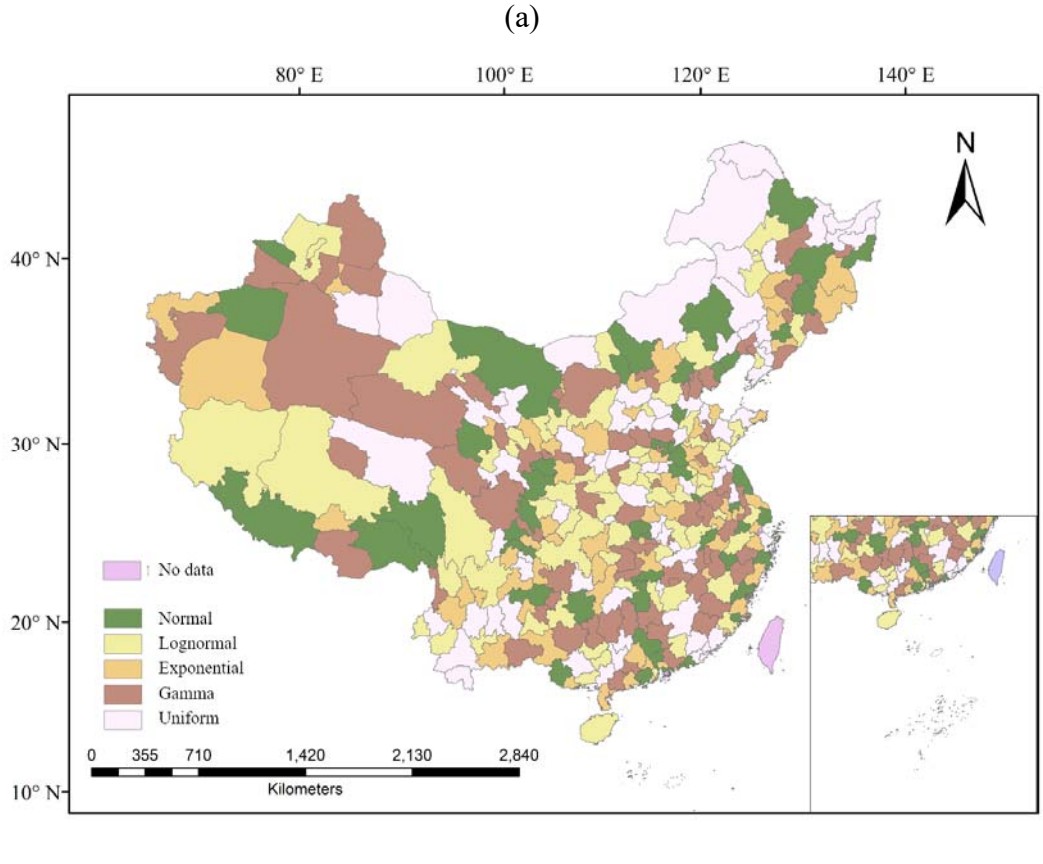


(b)

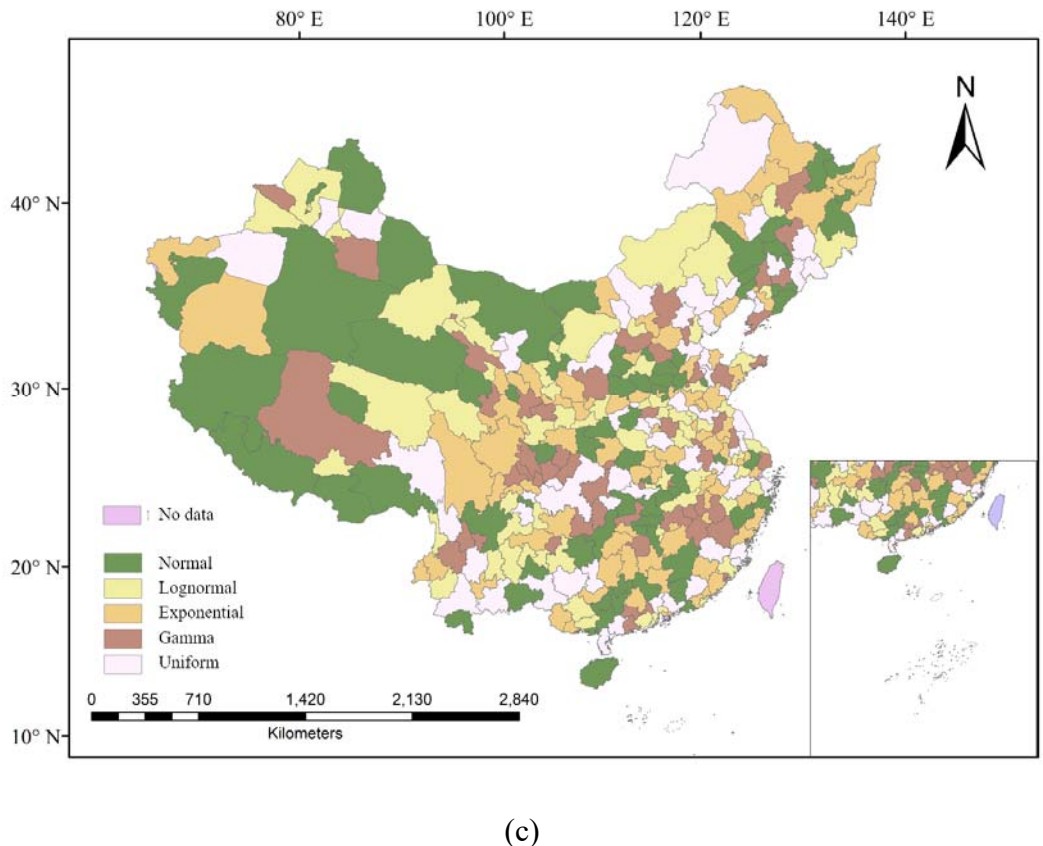


(c)
Figure. 4 Probability distribution types of water use at three spatial scales: (a) 1km scale; (b) appropriate
spatial scale; (c) prefecture scale
The optimal probability distributions of water use grid maps across three spatial scales (i.e., 1 km scale,
appropriate spatial scale, and prefecture scale) were identified and shown in Figure 4. The distribution types
of water use reflect the underlying dynamics of water use at various scales. At the 1 km scale (as shown in
Figure 4 (a)), most areas show a predominance of normal and log-normal distributions (i.e., in green and light
yellow), indicating more stable and symmetric water use patterns. Water use in these areas are consistent with
relatively socio-economic and climatic conditions, where are less fluctuated. However, the water use
distributions are more frequently exponential or gamma (i.e., in light orange and brown) in areas with rapidly
urbanization or industrialization in the central and eastern parts of China. And thus, water use are higher
variability and stochasticity. The distribution types of water use at the appropriate spatial scale (as shown in
Figure 4 (b)) are similar with those at 1 km scale as shown in Figure 4 (a). More exponential and gamma

distributions can be found in the significant agricultural activity areas at the appropriate spatial scale due to the irregular water use patterns driven by seasonal fluctuations and economic activities. When the spatial scale is up from 1 km to the appropriate ones, more nuanced understanding of water use dynamics can capture the impacts of local factors such as crop irrigation and industrial processes. Although the exponential distribution are found to be one of the prevalent types in regions with irregular or rapidly changing water use patterns at the prefecture scale as shown in Figure 4 (c), uniform and gamma distributions have appeared more frequently at this bigger prefecture scale. The transition to coarser spatial resolution leads to reduced spatial variability in water use, and the heterogeneity of water use is potentially oversimplified.

Based on the historical water use record from 1998 to 2013, a transition matrix for each grid cell was constructed to represent the probability of transition from its current state to the subsequent year. According to these transition matrices, the interval for the next state can be predicted, and a random sample is drawn from the corresponding probability distribution to obtain the simulated water use value. The probability rule is applied at three spatial scales (i.e., 1 km, appropriate scale, and prefecture scale). To improve clarity, only the simulated water use results for the validation period (2010–2013) are displayed in Figure 5, enabling a direct comparison of spatial patterns across scales during independent validation. The complete annual simulation results for the entire study period (1998–2013)—including all three spatial scales and both update rules—are available for download at: 10.6084/m9.figshare.30445157.

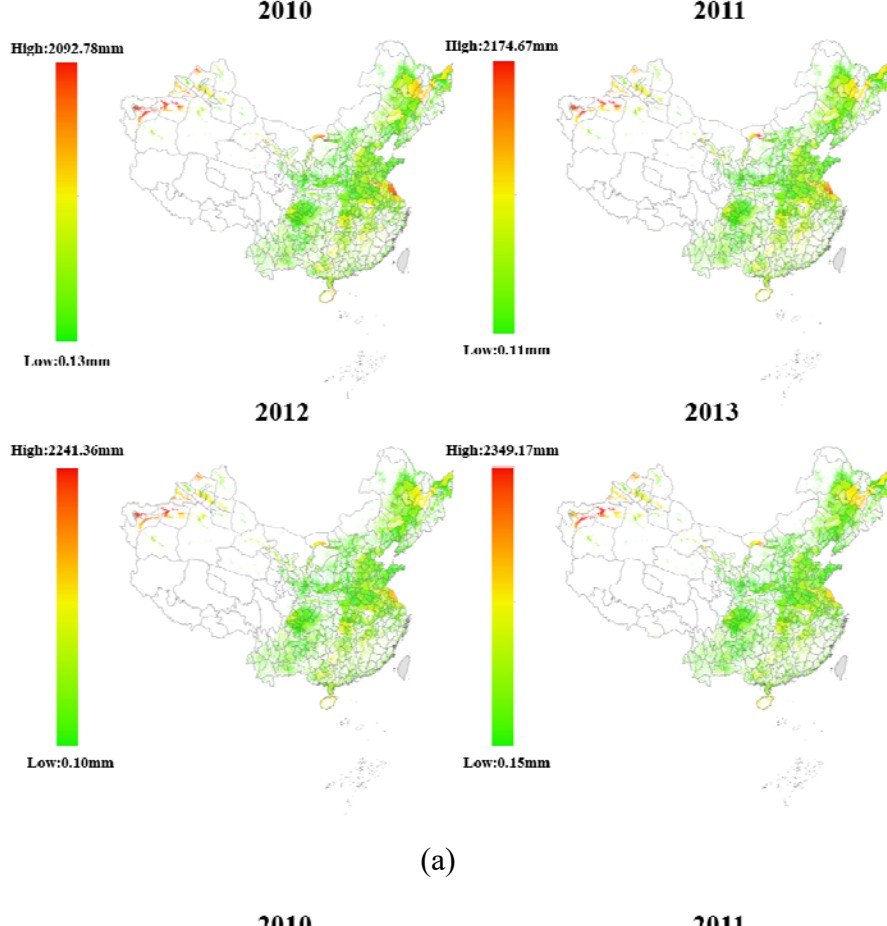

(a)

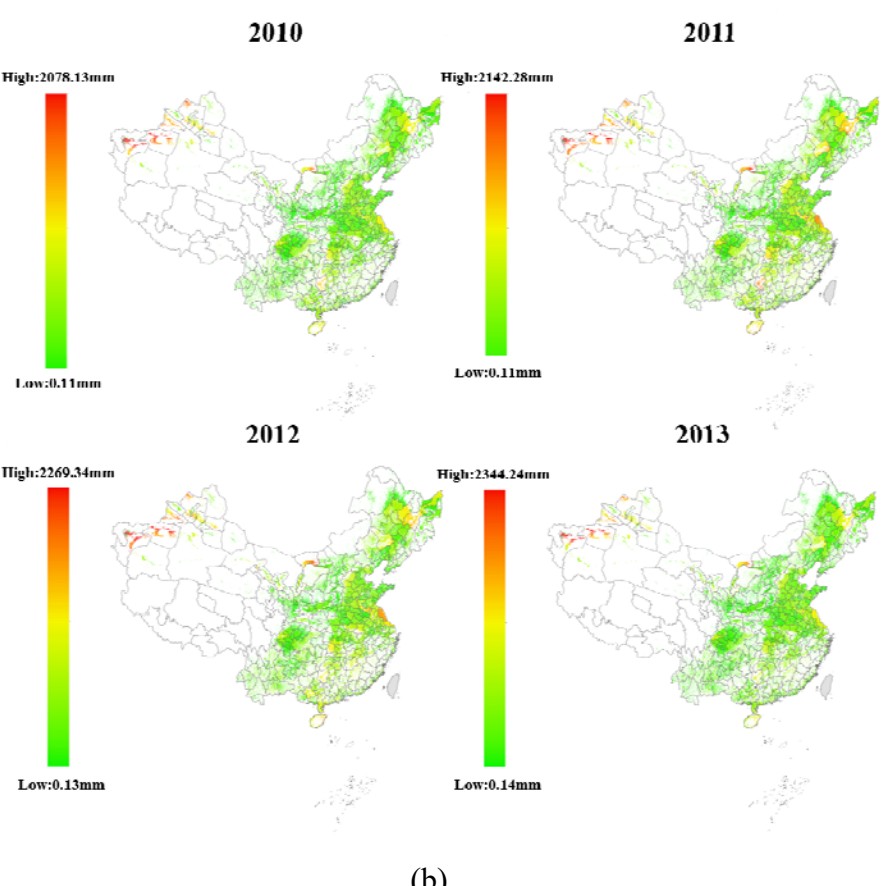

(b)

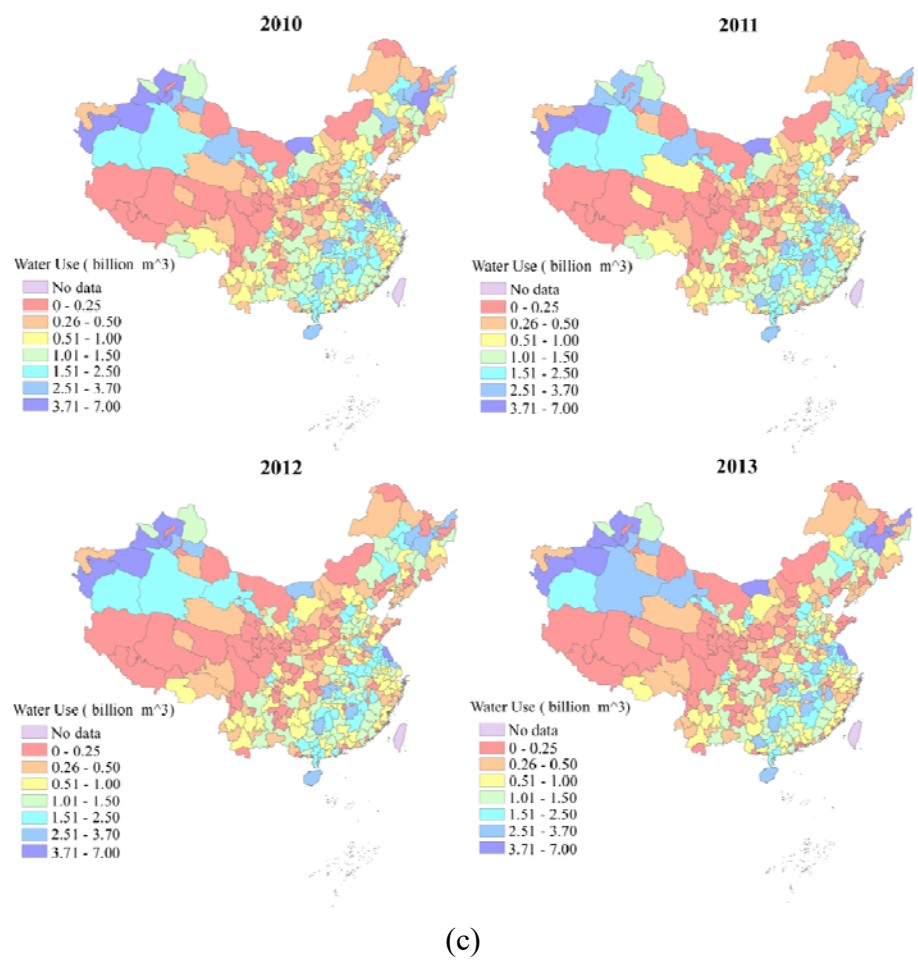

(c)

Figure. 5 Water use simulation results from the probability rule at three different scales: (a) 1km scale; (b)

appropriate spatial scale; (c) prefecture scale

Water use simulation maps from 1998 to 2013 through the probability rule, has clearly highlight the

spatial and temporal heterogenic at different spatial scales. The maps reveal the local increases in water use at
the 1 km scale (as shown in Figure 5 (a)), particularly in major metropolitan areas in eastern and central parts
of China. These increases can be found across the whole time of the selected series driven by the population
growth, industrial expansion, and urbanization. The results at the appropriate spatial scale (as shown in Figure
5 (b)) are different from those at the 1 km scale, their local variations become more homogeneous and
stationary. It demonstrates that the model can capture the macro-level variations across different geographical
areas. The water use trends are further found to be flatter with less spatial heterogeneity at the prefecture scale
(Figure 5 (c)). The local variations of water use cannot be reflected at the prefecture scale due to the coarser
spatial resolution. Areas with stable water use, such as the northern provinces, show a more uniform
distribution of water use, indicating that coarser resolution tend to mask these local variations.
The water use results at the three different spatial scales from probability rule show that the probability
rule CA model can effectively capture the spatial heterogeneity in water use. The incorporation of transition
probabilities and statistical distributions helps account for the temporal water use variations. However, water
uses in the areas with experiencing rapid changes show great fluctuations and spatial fragmentation, and they
are hard to be simulated by the probability rule. As the linear rule CA model can provide a more stable
alternative in rapid change areas, it can be implemented in water use simulation, too.

## 430 4.1.2 Water use simulation from linear rule CA model

There are three parameters to be calibrated in the linear rule CA model: the self-influence coefficient $\alpha$,
the neighboring influence coefficient $\beta$, and the spatial decay exponent $p$. The calibration and validation are
taken the statistics water use at the prefecture-level as the reference (i.e., observed water use data). Specifically,
for a given parameter set, the gridded water use is firstly simulated. The simulated grids are then aggregated
into the total water use at each prefecture scale along their boundaries. These total water uses are assessed by
the observed water use data from water resources bulletins and related statistical surveys. The calibration
period covers 1998–2009 and the parameter values are determined by minimizing *RMSE* and *RE* between the
simulated and observed total water use at the prefecture scale. The validation period covers 2010–2013 and
the performance of the model is also assessed by *RMSE* and *RE*. The optimal parameters at three spatial scales
during the calibration and validation periods, are illustrated in Figure 5.

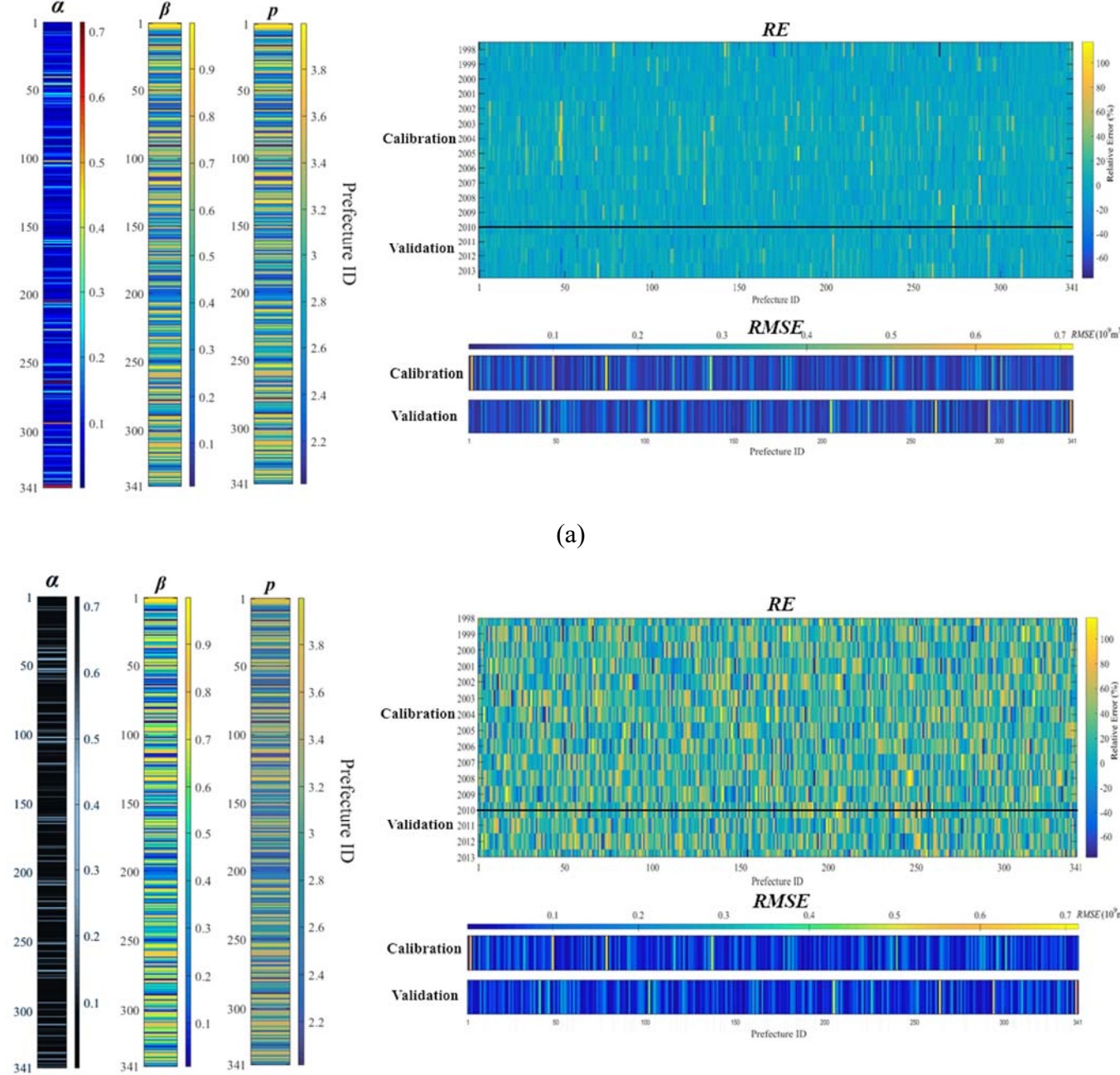

(a)

(b)

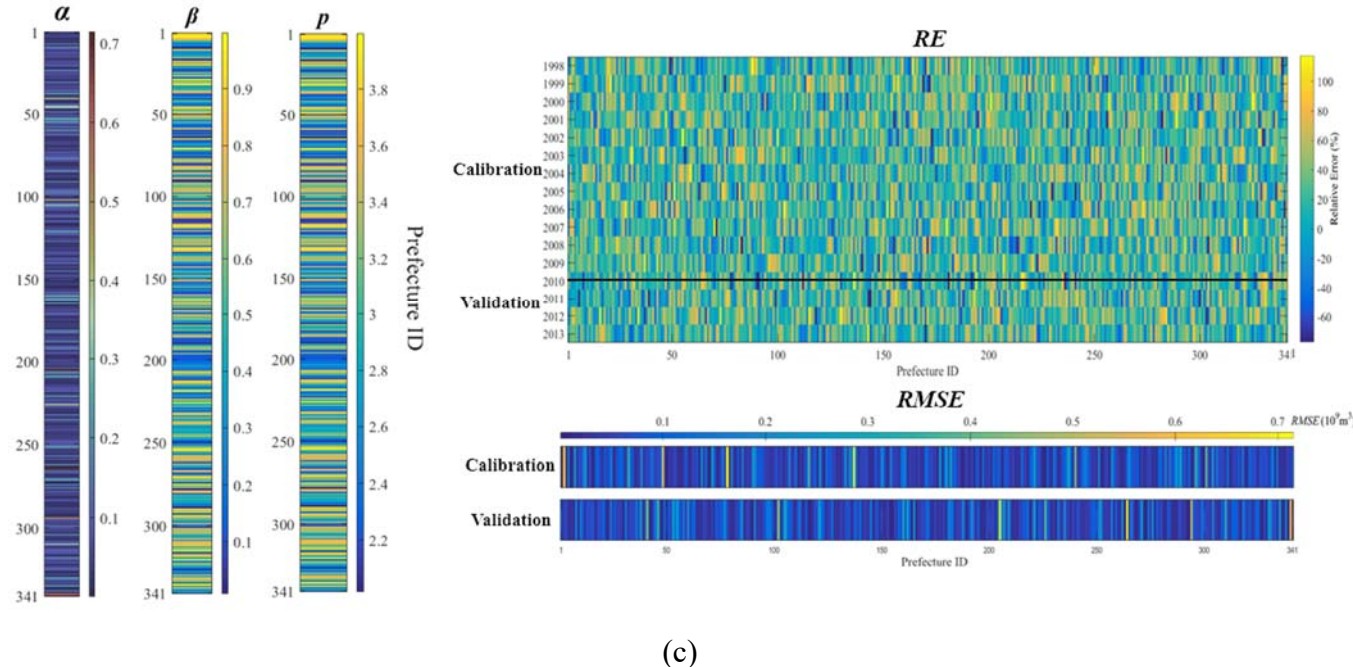


(c)

Figure 6. Optimal parameters of the linear rule CA model and the model performances at: (a) 1km scale; (b)

appropriate spatial scale; (c) prefecture scale

According to the results as shown in Figure 6, the calibrated parameters are varied with scales and the

performances of the models are acceptable. The spatial heterogeneity of the parameters $\alpha$, $\beta$, and $p$ tend to
reflect local water use patterns. There is a balance between $\alpha$ and $\beta$ in the areas with lower *RMSE* and *RE*
values from the results at 1 km scale as shown in Figure 5(a). And both self-dependence and spatial diffusion
jointly govern stable and reliable simulations. But at the appropriate spatial scale (Figure 6 (b)), the parameter
values show their deviations from those of the 1 km scale, and $\alpha$ and $\beta$ values become more regionally specific.
As $\beta$ plays a more dominant role in every prefecture, and higher *RMSE* and *RE* values are found. So only
heavily spatial diffusion may fail to capture the complexities of local water use dynamics. These areas often
experience a more varied water use. At the prefecture scale (Figure 6 (c)), the parameters are more likely to
be dominated by $\alpha$, especially in regions with large-scale infrastructure or agricultural variations. Their
simulation errors are higher because the spatial variations are more difficult to be captured at this prefecture
coarser resolutions. And the influence of $\beta$ tends to be lower at this scale.

After calibrating the parameters for the linear rule of the CA model, the water use grid maps were

generated at three spatial scales (i.e., 1 km, appropriate spatial scale, and prefecture scale). For brevity and
readability, Figure 7 presents only the simulation results for the validation years (2010–2013), which allow a
more straightforward assessment of the model's predictive performance and spatial differences among scales.
The full set of simulated water use maps from 1998–2013, can be accessed at: 10.6084/m9.figshare.30445157.

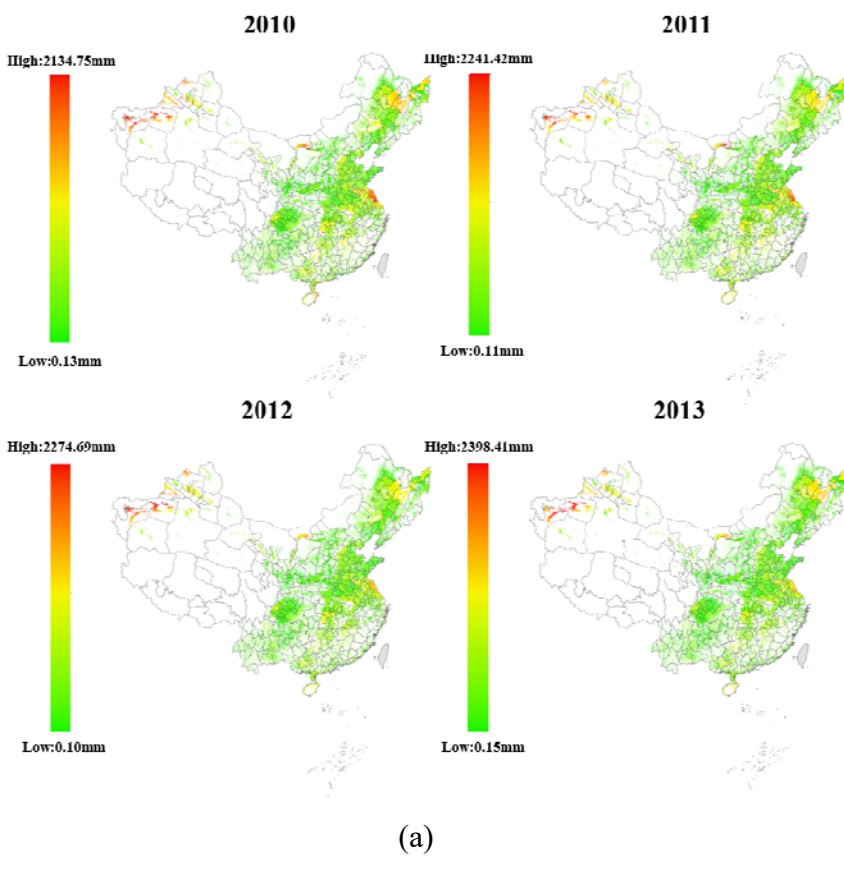

(a)

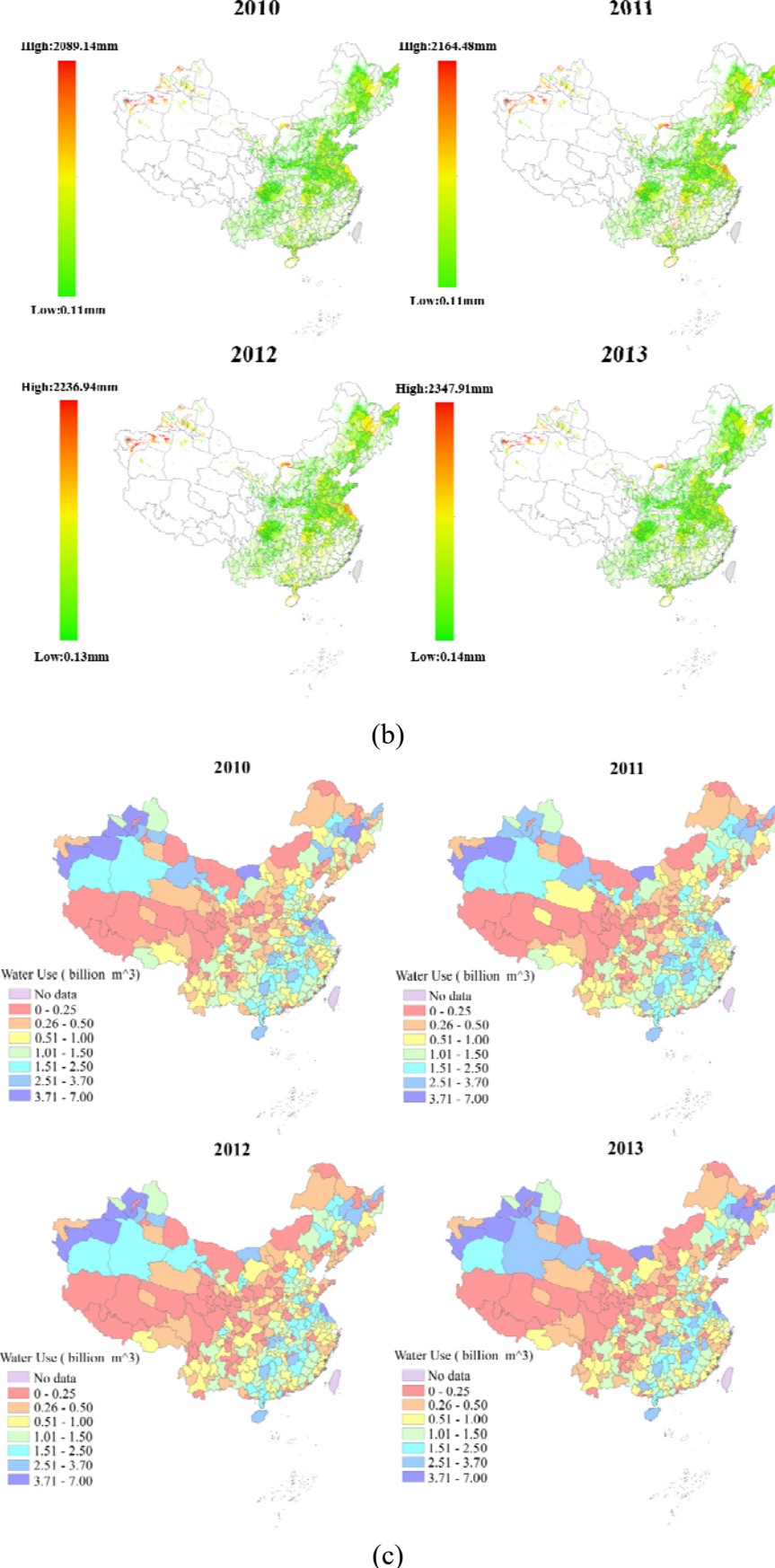

(b)

(c)

Figure 7. Water use simulation results from the linear rule at three different scales: (a) 1km scale; (b)

appropriate spatial scale; (c) prefecture scale

The simulated water use maps of the linear rule from 1998 to 2013 are shown in Figure 7, and they exhibit
clear spatial dynamics at different spatial scales. At the 1 km scale (Figure 7 (a)), the maps reveal the fine-
scale variations in water use, highlighting their localized hotspots of high-water use, particularly in
economically developed urban centers such as the Yangtze River Delta, the Pearl River Delta, and the Beijing–
Tianjin–Hebei region. There are upward trends, which aligns with the patterns of population growth, industrial
expansion, and urbanization. From the results at the appropriate spatial scale as shown in Figure 7 (b), the
localized variations in water use are captured with more clarity and precision. The appropriate scale provides
a balanced representation that captures both the fine-scale dynamics and broader regional trends. At the
prefecture scale (Figure 7 (c)), the water use patterns become more smoothed. This bigger resolution limits
the model's ability to capture the dynamics even though the general upward trend in water use in urbanized
regions is still observable. The localized fluctuations or abrupt changes driven by specific local policies or
external factors are not fully represented at these bigger scales.
Moderate water use levels have been found in the North China Plain (NCP), compared with northeastern
China, even though the NCP features intensive irrigation and high population density. Several factors
contribute to this pattern. First, the calibration and validation rely on prefecture-level statistical survey data
(e.g., water resources bulletins). In recent years, a decreasing trend of agricultural water use has been reported
in many NCP prefectures, partly due to improvements in irrigation efficiency, the implementation of water-
saving policies, and adjustments in cropping structures. Second, total water use in our study is an aggregate
of irrigation, domestic, and industrial sectors. While total water use in the NCP remains dominated by
irrigation, many northeastern prefectures show a growing contribution from industrial water use, particularly
heavy industries and thermal power generation, leading to higher overall totals. Third, climatic and water
availability differences also play a role. Despite a shorter growing season, northeastern China often supports

water-intensive crops and benefits from relatively abundant local water resources, which result in higher water use per unit area. In addition, some previous studies that reported higher water use in the NCP were sector-specific (mainly irrigation) or based on different temporal baselines, which partly explains the discrepancy with our aggregated results.

The simulation results across three different spatial scales can also indicate the impacts of scale on perceived water use. Finer-resolution maps reveal localized hotspots of high water use that can be averaged out in coarser resolution. It demonstrates that the CA model's flexibility and further confirms the importance of multi-scale analysis in understanding water use.

## 4.2 Uncertainty estimation of water use simulation from GLUE

To further assess the reliability and robustness of the water use simulations from the CA model, the uncertainty is quantified by GLUE. According to the calibrated parameters for the probability and linear rules, the uncertainty ranges of the model outputs across different spatial scales are obtained through GLUE. To figure out both the accuracy of simulations and the confidence levels of model predictions from the parameter uncertainties, ensembles of the parameters combination were generated by the uniform sampling as the description in Section 2.3. Corresponding results from the CA model were obtained and their performances are quantified by *RMSE* and *RE* metrics. If the *RMSE* and *RE* metrics are acceptable according to their pre-defined thresholds, the maximum and minimum of the simulation results from the CA model are taken as their uncertainty range. These uncertainties across the three spatial scales are shown in Figure 8 from the probability rule and in Figure 9 from the linear rule at the 95% confidence level (i.e., pre-defined threshold). These uncertainties are from the parameter variabilities. A wider range of the uncertainties indicates lower stable and reliable simulation results, while narrower ones suggest stable and reliable simulation results. The spatial variation of uncertainty ranges can also reveal the significant regional differences of the simulations at

different spatial resolutions.

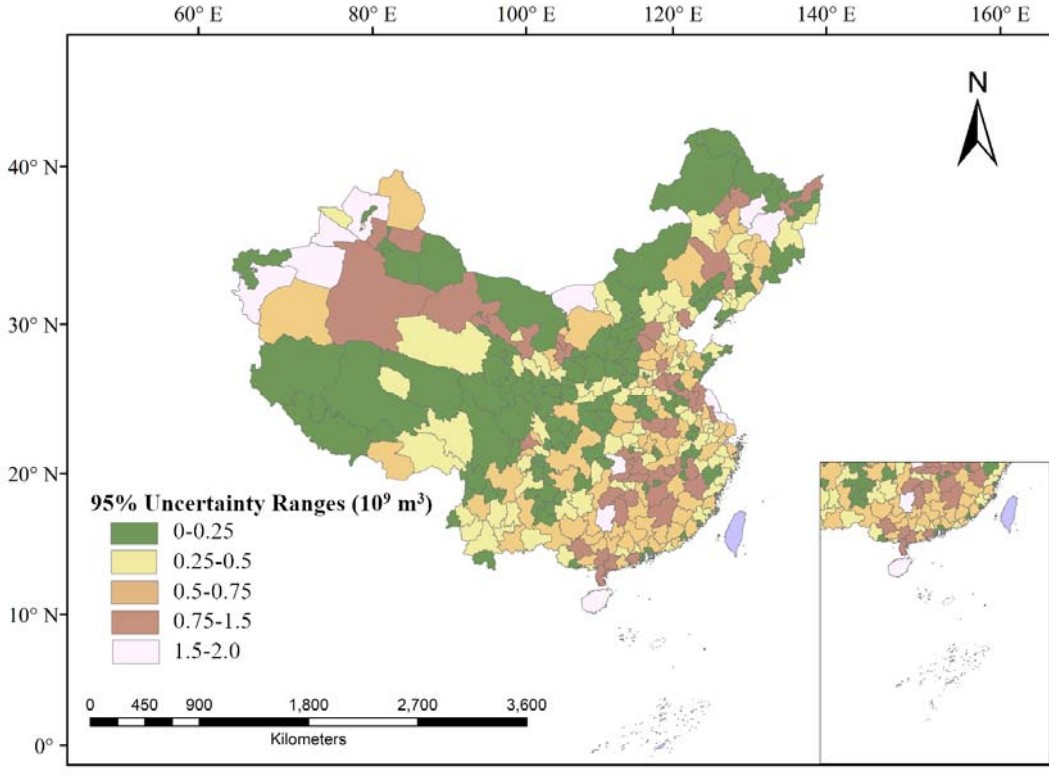

(a)

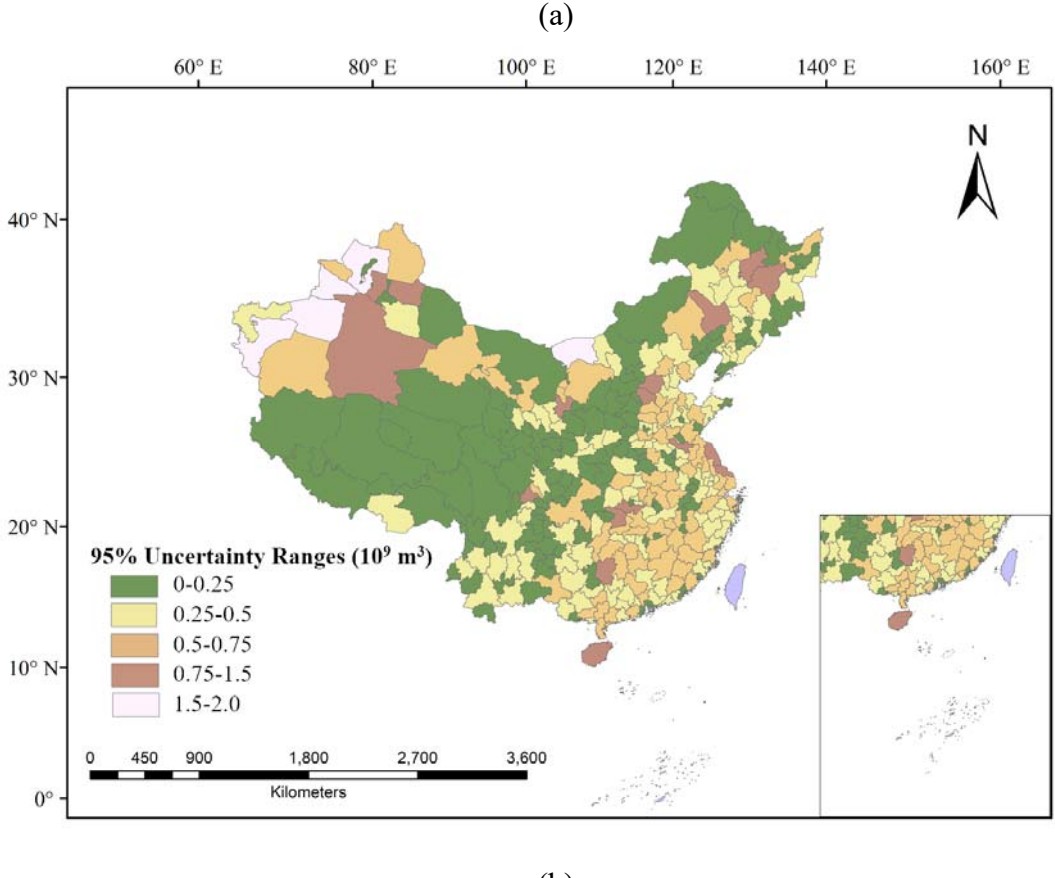

(b)

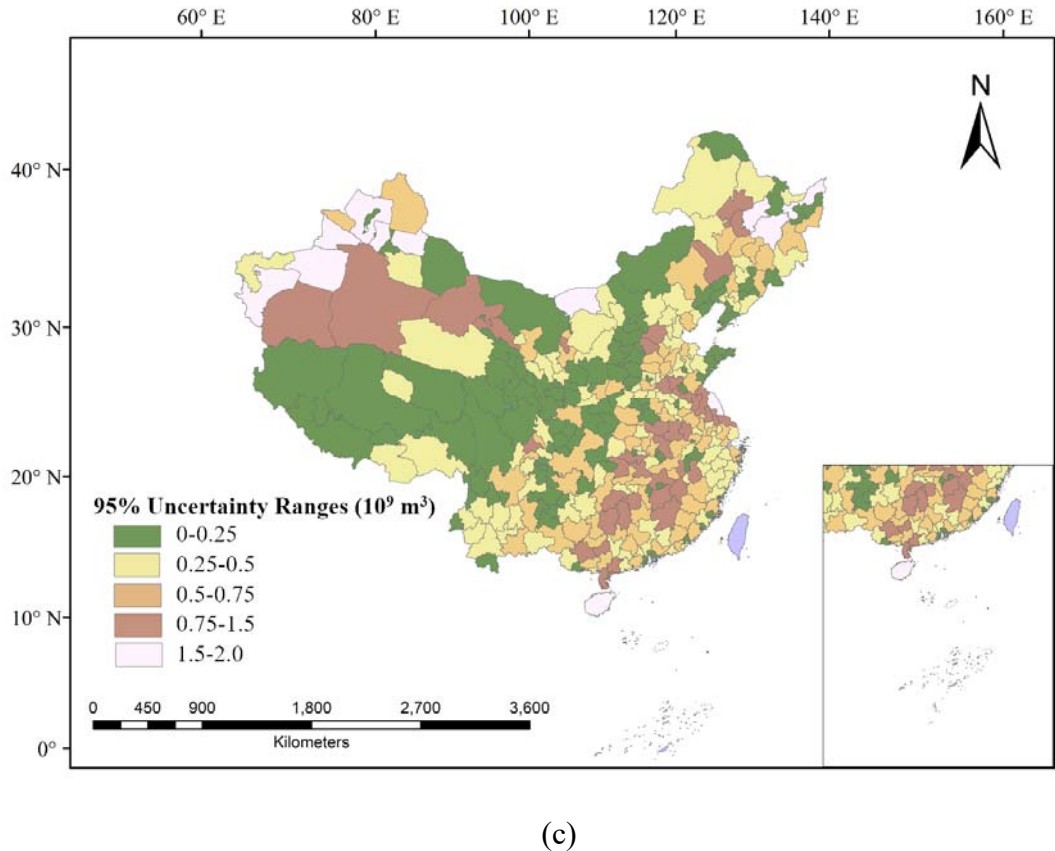

(c)

Figure 8. The uncertainty ranges of water use simulations at the 95% confidence level at three spatial scales

from the probability rule at: (a) 1km scale; (b) appropriate spatial scale; (c) prefecture scale

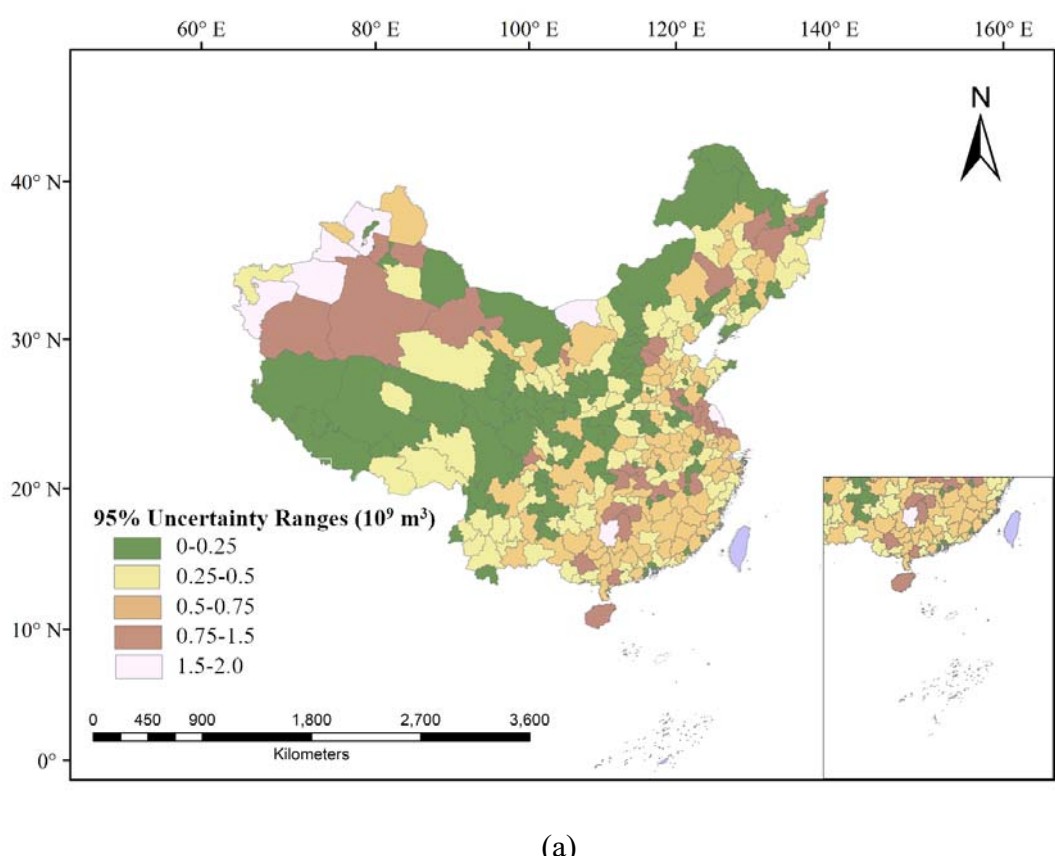

(a)

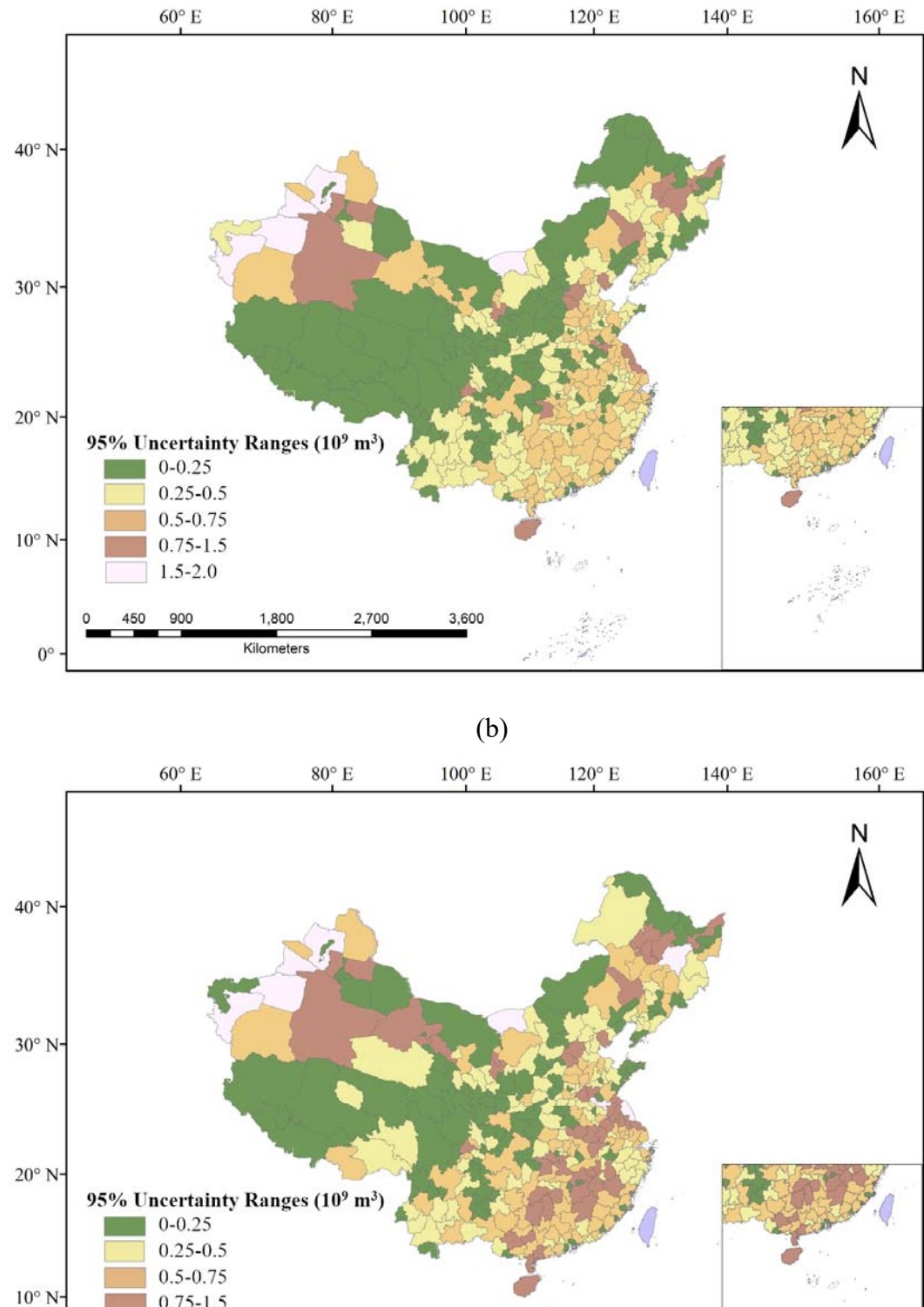


(b)


(c)

Figure 9. The uncertainty ranges of water use simulations at the 95% confidence level at three spatial scales

based on linear rule at: (a) 1km scale; (b) appropriate spatial scale; (c) prefecture scale

There are distinct regional patterns of simulation uncertainty from the spatial distribution of the uncertainty ranges at the 95% confidence level. Larger uncertainty ranges (i.e., $0.75$–$2.0*10^9 m^3$) are predominantly found in western and southwestern prefectures, such as Xinjiang, Qinghai, Gansu, and Chongqing, where data scarcity and complex local dynamics likely contribute to higher model uncertainty. In contrast, most eastern and northeastern regions, including Beijing, Jiangsu, Shandong, and Liaoning, exhibit relatively narrow uncertainty ranges (i.e., $0$–$0.5*10^9 m^3$), indicating more stable historical water use behavior and better alignment with CA model assumptions.

The model for the higher spatial resolutions has been found to be more sensitive to the local variations, and its uncertainty can be amplified in heterogeneous land use or socio-economic condition areas. For instance, large uncertainty ranges can be found in the densely populated urban centers or the regions undergoing rapid industrialization at the 1 km scale as shown in Figure 8 (a) and Figure 9 (a). But small uncertainty ranges can be found in the same areas from the coarser-scale simulations due to the averaging of the localized fluctuations as seen in Figure 8 (c) and Figure 9 (c) at the prefecture scale. However, the small uncertainty ranges are obtained at the cost of masking the sub-regional dynamics. There should be trade-off between capturing detail and maintaining stability. Thus, it is important to select an appropriate spatial scale for the specific planning or policy purposes.

The larger uncertainty ranges are more common from the probability rule than those from the linear rule, particularly in regions with complex or highly variable water use patterns. This is largely attributed to the probability rule itself, because there are randomness in both state transitions and value sampling in fitting distributions. Although the probability rule can help the model to capture non-linear dynamics and abrupt changes, it brings higher uncertainty into outputs. In contrast, the linear rule generally brings less uncertainty, reflecting the deterministic structure of its update mechanism in the model. The amount of water use in each

cell is predicted by the weighted influences from itself and neighboring cells. In regions with relatively stable
and spatially smooth development patterns, such as the eastern and northeastern parts of China, the linear rule
is more effective. However, the linear rule's assumption of gradual spatial continuity fails to capture abrupt
local changes in the rapidly changing or the weak spatial correlation water use areas—such as Fujian,
Chongqing, and some parts of Guizhou and Sichuan. And the linear rule potentially leads to an
underestimation of uncertainty.
To further examine the temporal dynamics and regional differences in model uncertainty, Daqing,
Chongqing, Fuzhou, Kashgar, Ningbo, and Bayannur are selected as six representative prefectures according
to their geographic locations, socio-economic conditions, and water use behaviors. The time series of 95%
uncertainty ranges from 1998 to 2013 for each representative prefecture is obtained from the results of the
probability and linear update rules at different spatial scales, illustrated as Figure 10.

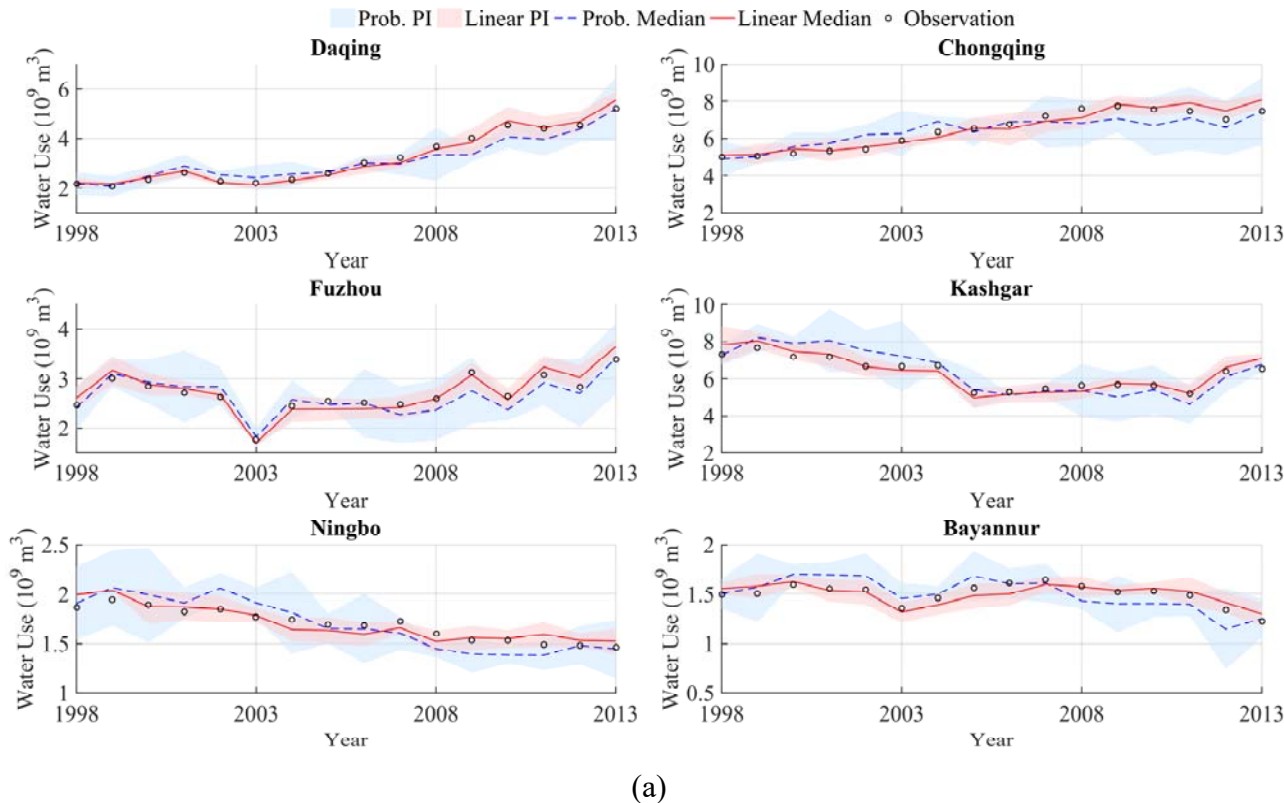


(a)

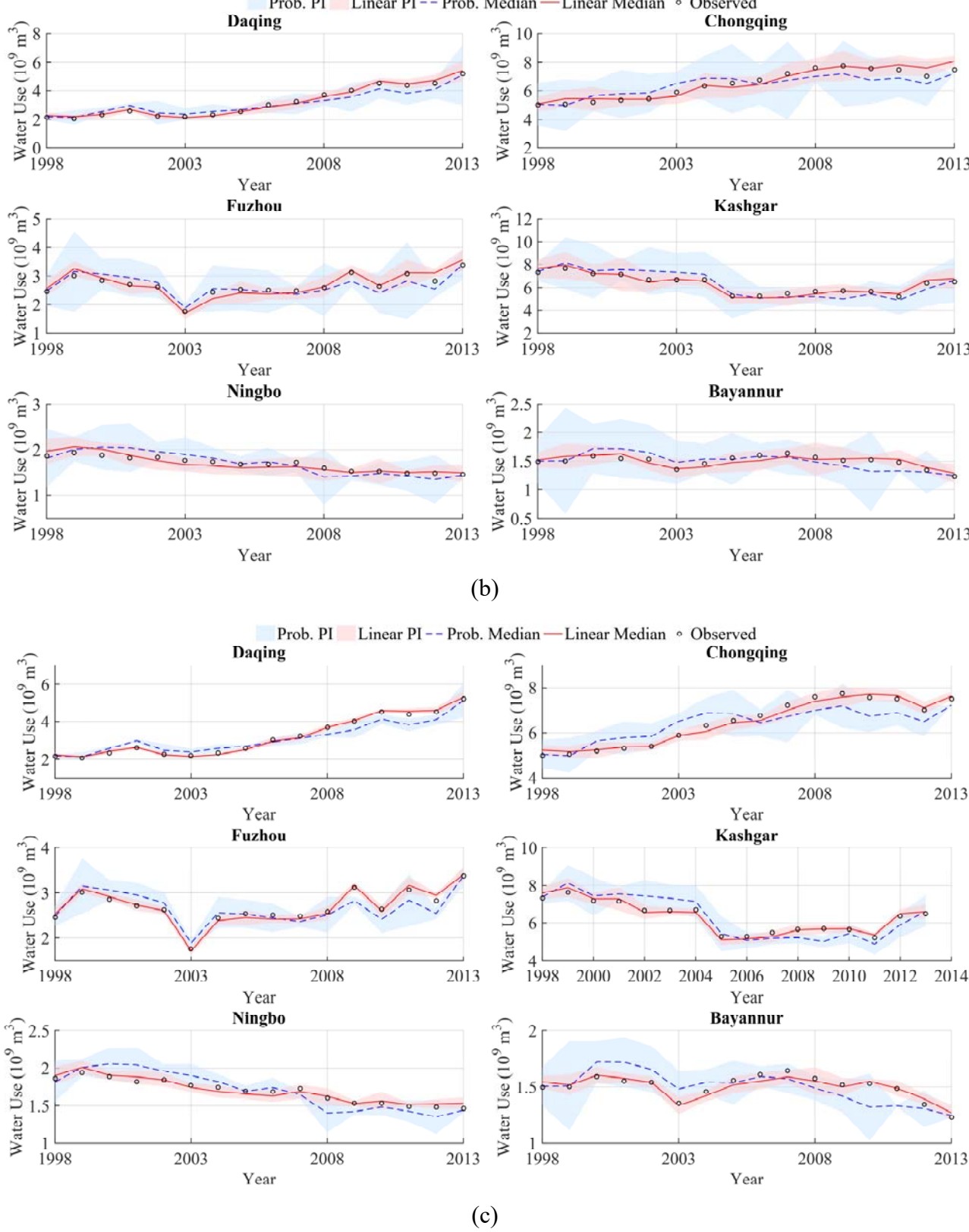

(b)

(c)

Figure 10. Time series of 95% uncertainty ranges from 1998 to 2013 for each representative prefecture from
both the probability and linear update rules at different spatial scales: (a) 1km scale; (b) appropriate spatial

scale; (c) prefecture scale

There is distinct the patterns across the selected prefectures as shown in Figure 10. The results from the
probability rule have been found to be consistently wider uncertainty intervals than that from the linear rule,
particularly in prefectures with more unstable water use conditions. For instance, Kashgar and Bayannur that
are located in arid and less-developed western regions exhibit larger and more variable uncertainty intervals
from the probability rule. And they are results of the combined effects of high inter-annual variability and
limited input data. In contrast, Ningbo and Daqing that are located in the more stable water use trends and
better data coverage areas have been found to show relatively narrow and consistent uncertainty from both
two rules. It is also interesting to find that the moderate uncertainty levels with obvious differences between
the two rules have been in Chongqing and Fuzhou with too many transitional urban areas. As the rapid
urbanization and economic shifts bring the localized instability, the probability rule can capture the shifts
whereas the linear rule can only underestimate uncertainty during periods of structural change due to its lower
sensitivity.
Beside the temporal and rule-based variability, the spatial scale size also plays a crucial role in shaping
simulation uncertainty. With the spatial resolution of the model increases, the simulation uncertainty tends to
increase, especially in regions with high spatial heterogeneity, such as arid or rapidly developing areas. The
finer resolutions can effectively capture the localized variations in water use, there are higher levels of
uncertainty due to the increased sensitivity to local fluctuations. In contrast, the results at coarser resolutions
tend to be smoothed these local variations, there will be narrower uncertainty but potentially overlooking the
sub-regional dynamics.
The uncertainty results have highlighted the importance of the spatial and temporal variations for
evaluating the simulation model performance and uncertainty. The results of the representative six prefectures
suggest that there is no one single update rule can outperform the others. Instead, the most appropriate rule
depends on the local water use dynamics and the purpose of the water use simulation.

## 5 Discussion

## 5.1 Impacts of Update Rules on Water Use Simulation

To assess the performance of the two update rules, the simulation results at three spatial scales were
upscaled to the provincial administrative level, and the *RMSE* and *RE* metrics were calculated for both the
probability and linear rules. An additional indicator, $\Delta signed$ was introduced to quantify both the magnitude
and direction of the differences between the two simulations (probability vs. linear). $\Delta signed$ represents the
mean signed grid-level difference normalized by the mean simulated water use for each province. A positive
$\Delta signed$ indicates that the probability rule yields higher simulated water use than the linear rule, while a
negative $\Delta signed$ indicates lower simulated values. There are notable differences between the results
produced by the two update rules, as summarized in Table 2.

Table 2. Model performance at the provincial administrative level from different update rules

| Provinces | *RMSE* (billion m$^3$) | | *RE* (%) | | $\Delta signed$ (%) |
| | Linear rule | Probability rule | Linear rule | Probability rule | |
| --- | --- | --- | --- | --- | --- |
| Beijing | 0.03 | 0.05 | +12 | +19 | +2.8 |
| Tianjin | 0.02 | 0.02 | -11 | -17 | -0.6 |
| Shanghai | 0.02 | 0.04 | -13 | -25 | +3.1 |
| Chongqing | 0.04 | 0.07 | +16 | +17 | +1.5 |
| Anhui | 0.12 | 0.17 | +23 | +32 | +2.2 |
| Fujian | 0.17 | 0.21 | -23 | +37 | +3.4 |
| Gansu | 0.35 | 0.43 | +27 | +29 | -1.9 |
| Guangdong | 0.26 | 0.36 | +21 | +32 | +3.9 |
| Guizhou | 0.31 | 0.40 | -26 | -34 | -1.2 |
| Hainan | 0.11 | 0.21 | -12 | +19 | +3.0 |

| Heilongjiang | 0.42 | 0.37 | -27 | +24 | +2.1 |
|---|---|---|---|---|---|
| Hunan | 0.21 | 0.11 | -16 | +10 | +1.6 |
| Jilin | 0.36 | 0.43 | +28 | +25 | +2.5 |
| Jiangsu | 0.12 | 0.17 | -12 | -26 | -0.7 |
| Jiangxi | 0.22 | 0.26 | +24 | -31 | +1.3 |
| Inner Mongolia | 0.19 | 0.29 | -23 | -35 | -2.1 |
| Qinghai | 0.31 | 0.41 | +43 | +36 | +1.4 |
| Ningxia | 0.11 | 0.20 | -31 | -41 | -2.4 |
| Shandong | 0.21 | 0.36 | +29 | -36 | +4.1 |
| Shanxi | 0.42 | 0.61 | +31 | +42 | +3.3 |
| Shannxi | 0.27 | 0.43 | -32 | -39 | -1.5 |
| Sichuan | 0.39 | 0.41 | +31 | +46 | +2.7 |
| Xizang | 0.56 | 0.61 | -62 | -79 | -0.9 |
| Xinjiang | 0.68 | 0.72 | +69 | +83 | +3.5 |
| Yunnan | 0.25 | 0.34 | -37 | -47 | -1.1 |
| Zhejiang | 0.21 | 0.29 | +18 | +12 | +1.9 |
| Guangxi | 0.34 | 0.42 | -32 | -39 | -1.7 |
| Hubei | 0.29 | 0.39 | +27 | +26 | +2.2 |
| Liaoning | 0.35 | 0.41 | +33 | +48 | +3.7 |

The linear rule generally exhibits lower *RMSE* and *RE* values, indicating higher stability and consistency
with observed prefecture-level statistics. This is because the linear rule updates each grid deterministically
based on spatial averages of its neighbors, which smooths fluctuations and captures persistent long-term
patterns. Consequently, it tends to reduce local variability and enhance regional stability, especially at coarser
spatial scales. By contrast, the probability rule explicitly incorporates stochasticity through distribution-based
state transitions. This enables it to capture local irregularities, abrupt changes, and nonlinear water use
dynamics driven by variations in industrial structure, irrigation demand, or climatic conditions. However, such
stochastic behavior can also amplify uncertainty in regions with sparse data or complex spatial heterogeneity,
resulting in slightly higher *RMSE* and *RE* values.
Quantitatively, the mean *RMSE* of the linear rule during the validation period (2010–2013) was 0.28
billion m³, compared with 0.36 billion m³ for the probability rule. The corresponding mean relative errors
were ±22.4% and ±29.8%, respectively. At the national scale, the simulated total water use was 570.6 billion
m³ for the linear rule and 583.2 billion m³ for the probability rule, differing by +1.1% and +3.5% from observed
national statistics. Therefore, the linear rule is identified as the best-performing estimation framework for
reproducing the observed spatiotemporal water use distribution in China, whereas the probability rule provides
valuable complementary insights for representing uncertainty and local heterogeneity.
The $\Delta signed$ results indicate that the probability rule tends to simulate higher water use in industrially
intensive provinces (e.g., Shandong, Guangdong, and Liaoning) and slightly lower values in water-scarce
inland regions (e.g., Ningxia, Gansu, and Inner Mongolia). The results confirm that the two rules emphasize
different aspects of the underlying processes—the probability rule better reflects stochastic local variability,
while the linear rule offers greater stability and smoother large-scale consistency.

## 632  5.2 Impact of Spatial Scales on Water Use Simulation

To investigate how spatial resolution influences the accuracy of the water use simulation, their
performances at the three spatial scales (i.e., 1 km scale, appropriate intermediate scale, and prefecture scale)
are evaluated and are aggregated to the provincial level to ensure comparability across scales. Their results
are summarized in Table 3 and indicate their notable differences in simulating accuracy depending on its
spatial scales.
Table 3. Model performance at the provincial administrative level at different spatial scales

| Provinces | *RMSE* (billion m³) | | | *RE* (%) | | |
|---|---|---|---|---|---|---|
| | 1 km scale | Appropriate | Prefecture | 1 km scale | Appropriate | Prefecture |

|  |  | scale | scale |  | scale | scale |
| --- | --- | --- | --- | --- | --- | --- |
| Beijing | 0.04 | 0.03 | 0.05 | +22 | +12 | +19 |
| Tianjin | 0.04 | 0.02 | 0.06 | +26 | -11 | -17 |
| Shanghai | 0.03 | 0.02 | 0.04 | -23 | -13 | -25 |
| Chongqing | 0.06 | 0.04 | 0.07 | -26 | +16 | +27 |
| Anhui | 0.17 | 0.12 | 0.19 | +35 | +23 | +32 |
| Fujian | 0.23 | 0.17 | 0.21 | -31 | -23 | +37 |
| Gansu | 0.41 | 0.35 | 0.43 | +31 | +27 | +29 |
| Guangdong | 0.31 | 0.26 | 0.36 | -29 | +21 | +32 |
| Guizhou | 0.36 | 0.31 | 0.40 | +31 | -26 | -34 |
| Hainan | 0.19 | 0.11 | 0.21 | +16 | -12 | +19 |
| Heilongjiang | 0.49 | 0.42 | 0.47 | +31 | -27 | +34 |
| Hunan | 0.26 | 0.21 | 0.31 | -19 | -16 | +20 |
| Jilin | 0.42 | 0.36 | 0.43 | -31 | +28 | +35 |
| Jiangsu | 0.19 | 0.12 | 0.17 | -21 | -12 | -26 |
| Jiangxi | 0.31 | 0.22 | 0.36 | -26 | +24 | -31 |
| Inner Mongolia | 0.26 | 0.19 | 0.29 | -31 | -23 | -35 |
| Qinghai | 0.40 | 0.31 | 0.41 | +52 | +43 | +56 |
| Ningxia | 0.21 | 0.11 | 0.30 | -39 | -31 | -41 |
| Shandong | 0.29 | 0.21 | 0.36 | -32 | +29 | -36 |
| Shanxi | 0.53 | 0.42 | 0.61 | +36 | +31 | +42 |
| Shannxi | 0.34 | 0.27 | 0.43 | -36 | -32 | -39 |
| Sichuan | 0.42 | 0.39 | 0.41 | +41 | +31 | +46 |
| Xizang | 0.67 | 0.56 | 0.61 | -74 | -62 | -79 |
| Xinjiang | 0.79 | 0.68 | 0.72 | +72 | +69 | +83 |
| Yunnan | 0.31 | 0.25 | 0.34 | -49 | -37 | -47 |
| Zhejiang | 0.24 | 0.21 | 0.29 | +23 | +18 | +32 |
| Guangxi | 0.39 | 0.34 | 0.42 | -36 | -32 | -39 |
| Hubei | 0.35 | 0.29 | 0.39 | +31 | +27 | +36 |
| Liaoning | 0.37 | 0.35 | 0.41 | +42 | +33 | +48 |

The relatively lower accuracy of the 1 km scale simulations is attributed to the increased sensitivity to local heterogeneity. At the 1km scale, small errors from the input variables or local noise can accumulate and amplify to larger discrepancies. Although the results from1kmscale take more spatial detail to reflect the variations in water use, it also brings the uncertainty, especially in the regions with the fragmented data or the high socio-economic diversity. The simulation results from the prefecture level tend to oversimplify the spatial variation. Generally, the results from the coarse resolution have smoothed the intra-regional differences in water use patterns, and resulted in under or over-predictions at the provincial administrative level. The reduced spatial granularity leads to reduction in uncertainty of simulation but can mask the disparities that are critical to policy implementation and resource allocation.

The results from the appropriate scale can balance the spatial sensitivity and model stability. It is fine enough to capture meaningful spatial heterogeneity, and it is yet coarse enough to mitigate excessive noise and data sparsity effects. The most reliable simulating water use should be based on this scale. And it should be noted that that it is crucial for improving simulation reliability to select an appropriate spatial resolution that aligns with both the model structure and the scale of decision-making.

## 5.3 Spatial Heterogeneity of Water Use Grid Maps

To better understand the spatial heterogeneity of simulated water use, the Coefficient of Variation (CV) and Moran's I were applied to quantify the variability and spatial autocorrelation of water use across three spatial scales — 1 km, the appropriate spatial scale, and the prefecture scale. These two metrics together reveal how spatial scale and model design influence the representation of water-use heterogeneity.

At the 1 km scale, the highest CV values are observed among the three scales, indicating the greatest variability in water use across grid cells. This fine resolution captures the most detailed local differences, especially in areas with intensive agricultural or industrial activities. Moran's I results also reveal strong

positive spatial autocorrelation, suggesting distinct clustering of high- or low-water-use regions, particularly
around large urban centers and irrigation districts.
At the appropriate spatial scale, both the CV and Moran's I values indicate a moderated heterogeneity
pattern. Compared with the 1 km scale, the variability decreases as small-scale noise is smoothed out, while
the spatial clustering remains evident but less fragmented. This scale provides a balanced representation by
capturing regional heterogeneity without introducing excessive spatial detail or instability. The appropriate
spatial scale therefore achieves the optimal trade-off between capturing local patterns and maintaining spatial
coherence.
At the prefecture scale, the lowest CV values are recorded, showing that much of the local variability has
been smoothed. Moran's I values remain moderately positive, reflecting that some regional clustering persists
but overall spatial dependence becomes less pronounced. At this coarser scale, water-use patterns become
generalized, reducing the granularity of spatial differences.
To further quantify these relationships, Table 4 presents the average CV and Moran's I values under
different spatial scales and update rules.
Table 4 Comparison of CV and Moran's I under different spatial scales and update rules

| Spatial scale | Update rule | *CV* (mean) | Moran's I (mean) | Interpretation |
|---|---|---|---|---|
| 1 km | Probability rule | 0.82 | 0.59 | Highest heterogeneity; strong clustering in high-use regions. |
| 1 km | Linear rule | 0.78 | 0.56 | Slightly smoother than the probability rule; still high local variation. |
| Appropriate scale | Probability rule | 0.71 | 0.72 | Balanced heterogeneity and spatial dependence; optimal trade-off between detail and stability. |
| Appropriate scale | Linear rule | 0.68 | 0.70 | Slightly lower variability, indicating smoother transitions among neighboring cells. |
| Prefecture scale | Probability rule | 0.60 | 0.74 | Variability largely smoothed; moderate clustering remains. |
| Prefecture scale | Linear rule | 0.57 | 0.73 | Lowest heterogeneity; strong regional aggregation of water-use patterns. |

In addition to the influence of spatial scale, the choice of update rule also affects the spatial heterogeneity

of simulated water use. Across all scales, the probability rule yields slightly higher CV values than the linear rule, indicating that it better preserves local variability and stochastic fluctuations of water use. This difference stems from the probabilistic state-transition mechanism, which allows grid-level water use to fluctuate around its expected trajectory, capturing year-to-year uncertainty that is smoothed out in the deterministic linear formulation.

Conversely, Moran's I values under the linear rule are comparable to or slightly lower than those from the probability rule, suggesting smoother and more spatially continuous water-use patterns. This indicates a stronger spatial-averaging effect of the deterministic update mechanism, which may be advantageous for long-term or regionally aggregated assessments but less effective in representing local variability.

Overall, both spatial scale and update rule jointly shape the representation of spatial heterogeneity in water-use simulation. The 1 km scale captures the most detailed variability, the appropriate spatial scale provides a balanced and realistic depiction of regional patterns, and the prefecture scale generalizes spatial differences. Meanwhile, the probability rule emphasizes local randomness and uncertainty, whereas the linear rule accentuates deterministic spatial continuity.

There had been some water use simulation results at previous studies. Fox example, Huang et al. (2018) produced a global-scale monthly water withdrawal dataset at 0.5° resolution, distinguishing six sectors (e.g., irrigation, domestic, electricity generation, livestock, mining, manufacturing) over the period 1971–2010; Hou et al. (2024) developed China's industrial water withdrawal dataset (CIWW), providing gridded monthly data from 1965 to 2020 at 0.1° and 0.25° resolutions; Zhang et al. (2025) presented a high-resolution sectoral water use dataset (HSWUD) for mainland China, covering irrigation, manufacturing, thermal power cooling, and domestic use at 0.1° × 0.1° resolution, with strong consistency to prefecture-level statistics ($R^2 \approx 0.88$). As the results shown in the previous sections, our dataset is generated at different spatial resolutions (e.g., 1 km×1

699 km, appropriate spatial scale), enabling detailed representation of spatial heterogeneity within prefectures.

700 Due to the substantial differences in spatial resolutions between these datasets, it is not easy to compare the

701 differences of the spatial distribution patterns. But the relative values of performance metrics such as *RMSE*

702 and *RE* can figure out the better one among them. The values of *RMSE* within 0.1 (i.e., normalized by mean

703 water use) and a *RE* within −20 % to +30 % are found across all prefectures according to the results of our

704 simulation. And all these results are consistent.

705  Thus, the results show that, relative to the three reference datasets, our model's prefecture-level water

706 use estimates achieve a *RMSE* within 0.1 (normalized by mean water use) and a *RE* within −20 % to +30 %

707 across all prefectures. These results within the range generally are considered acceptable for large-scale water

708 use modeling, indicating that our estimates are consistent with these previous studies while offering finer

709 spatial details.

710 **6 Conclusion**

711  A multi-scale water use simulation framework has been proposed through integrating a CA model with

712 GLUE to address spatial heterogeneity and uncertainty in this study. Both probability rule and linear rule

713 across three spatial scales (i.e., 1 km, appropriate scale, and prefecture scale) have been applied over 341

714 prefectures in China as a case study. The impacts of model structure and spatial scale on the spatial

715 heterogeneity and uncertainty have been figured out.

716  It is interesting to find that the probability rule effectively captures stochastic variations and abrupt

717 transitions in water use but brings larger uncertainty due to its random sampling nature. And the linear rule

718 brings more stable and accurate simulations, particularly in regions with smoother water use patterns. The

719 local noise and uncertainty tend to be amplified in the results of the water use simulation at the 1km scale ,

720 while the essential spatial heterogeneity tend to be oversimplified and suppressed in the results at the

prefecture scale The most reliable simulation are found from the appropriate scale due its trade-off between
capturing spatial heterogeneity and maintaining model stability.

Future improvements for the water use simulation should involve the adaptive update rules that respond

to external drivers such as policy shifts or climate shocks, and extend the simulation to finer temporal scales
(e.g., seasonal or monthly) for improved short-term decision making. More uncertainty quantification methods,
such as Bayesian inference or Markov Chain Monte Carlo, are recommended to enhance performance in high-
dimensional settings. Overall, our study can contribute to the water use simulation at a multi-scale and with
uncertainty-aware. Our proposed framework will be helpful for the integrated water management,
infrastructure planning, and environmental policy under changing socio-economic and climatic conditions.

# Author contributions

JZ designed the model architecture, performed the computations, conducted the statistical analysis, and drafted the manuscript. DL acquired funding, contributed to the study design, provided research data, supervised the project, and guided the manuscript revision. JW contributed to manuscript revision discussions and provided advice on submission procedures. FU, LX, ZP, and WG participated in revision discussions and contributed to figure and chart preparation. All authors reviewed and approved the final version of the manuscript for submission.

# Competing interests

The authors declare that they have no conflict of interest.

# Acknowledgement

The authors gratefully acknowledge the financial support from National Key R&D Program of China (2024YFC3012402 and 2022YFC3202803), the National Natural Science Foundation of China (Nos.52379022, and 51879194).

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
