# Peer review of "Abstract: Reliable water use simulation is essential for sustainable water resource planning, especially under intensifying pressures from climate change, population growth, and socio-economic transitions. While previous studies have extensively explored water availability as supply side modelin"

_EGUsphere, 2025_

## Author Comment (AC1)

**Reference Number: egusphere-2025-2734**

**RESPONSES TO REVIEWER ONE'S COMMENTS**

We would like to express our sincere appreciation for your professional and insightful remarks on our paper. The comments are valuable and helpful for us to improve the quality of the manuscript. All the concerns raised have been carefully treated and an itemized reply to the reviewer's comments is presented in the revision files.

**COMMENT* 1:**

The main objective of this study is to address the impact of spatial scale on the spatial heterogeneity of water use. However, this study treats water use as a whole, without separating the spatial heterogeneity of water use by different sectors (e.g., irrigation, industrial and domestic). Actually, water use by different sectors shows large spatial variation which I think is important in water use modelling.

**COMMENT* 2:**

In the model framework, water use grid maps at different spatial scales are first prepared as inputs to the CA model. To generate the water use grid maps, the water use data at administrative survey scale is processed and downscaled to grid-based formats. Here, the iterative input selection algorithm is used to select the most relevant variables for water use, while the CNN model captures the relationships between input variables and water use. This step is important for the model performance, So I am wondering what are the most relevant variables for sectoral water use? This result should appear in this paper. Commonly, irrigation water use is mostly related to irrigated cropland area, and industrial/domestic water use are relevant to GDP/population density, as well as night light intensity. Whether the result of this study aligns with previous results?

**COMMENT* 3:**

L145: "the appropriate spatial scale for water use simulation is identified using an end-to-end deep learning-based spatial scale adaptive selection model". What is the definition of "appropriate spatial scale"? In my view, the appropriate spatial scale should be a clear spatial resolution (e.g., 5km, 10km), and may vary across water use sectors or limited water use dataset. It is suggested to clarify the result of the appropriate spatial scale in gridded water use simulation.

**RESPONSE to COMMENTS* 1-3:**

We appreciate the reviewer's comments regarding the treatment of different water use sectors, the selection of relevant variables, and the explanation of the appropriate spatial scale. In the revised manuscript, we have substantially expanded Section 2.1 to clarify these points. In preparing the water use grid maps, we first generated separate

maps for irrigation, domestic, and industrial water use, based on prefecture-level statistical survey data and sector-specific predictor variables. These sectoral grids were then aggregated into total water use maps for modeling. This choice is based on two main considerations: (1) during the study period (1998–2013), China's total water use at the national scale entered a relatively stable stage in its temporal trend, with an average annual growth rate of only about 0.87% (from 505.53 billion $m^3$ in 1998 to 575.44 billion $m^3$ in 2013), mainly due to policy interventions, technological improvements, and changes in industrial structure. Since the primary objective of this study is to investigate the effects of spatial scale on the spatial heterogeneity of water use, it is more appropriate to adopt a temporally stable water use indicator as the simulation target, minimizing the confounding effects of sector-specific temporal fluctuations.

We also now explicitly list in Section 2.1 the most relevant input variables for each sector identified through the iterative input selection algorithm. These variables serve as inputs to the subsequent Convolutional Neural Network (CNN) model for downscaling prefecture-level data to grid-based water use maps.

The "appropriate spatial scale" in this study is a concept proposed in our previous work, designed to balance simulation accuracy with the spatial information density of rasterized water use data. In earlier studies, fixed spatial scales were applied across different prefectures, which failed to reflect the variability in land area, natural endowments, and water use structures among cities. This often led to discrepancies in information density across the simulated rasters. Excessively high resolutions could cause over- or underestimation due to data limitations, whereas overly coarse resolutions could obscure critical spatial variations. To address this, we previously developed a deep learning-based spatiotemporal scale adaptive selection (SSAS) model, in which the spatial scale selection module identifies the optimal spatial resolution of input variables by maximizing information density, quantified using Conditional Entropy, while balancing simulation accuracy through Kullback-Leibler Divergence Loss and Relative Error. This approach enables each prefecture to have its own optimal spatial scale rather than adopting a uniform resolution.

In the revised manuscript, we have substantially expanded Section 2.1 ("Water Use Grid Maps Generating") to address these points. The revised section now reads:

**Revised Section 2.1:**

The spatial scale of water use simulation is determined by the spatial scale of the input data, so water use grid maps at different spatial scales were prepared as input to the simulation model. To obtain the water use grid maps, several steps should be done to convert the water use data at administrative survey scale into spatially explicit grids of varying resolutions.

The grid maps of irrigation, domestic, and industrial water use are generated from the prefecture-level statistical survey data and water use sector-specific predictor variables. For each sector, the most relevant input variables are identified through an iterative input variables selection algorithm (Zhang et al., 2023; Zhang et al. 2025). Specifically, irrigation water use was modeled by the potential evapotranspiration, normalized difference vegetation index (NDVI), rainfall and soil moisture; domestic

water use was modeled by population, rainfall, temperature and night-light; industrial water use was modeled by GDP, night-light, population and rainfall. And then these sectoral gird maps were aggregated to form total water use grid maps for modeling. This aggregation is done for two reasons: the first one is that the temporal trend of the total water use has become stable during the study period (1998–2013) and future. The average annual growth rate is only about 0.87% (from 505.53 billion m3 in 1998 to 575.44 billion m3 in 2013) due to the policy interventions, technological improvements, and industrial structure changes. Since the primary objective of our study is to examine the influence of spatial scale on the spatial heterogeneity of water use, a temporally stable indicator helps minimize the confounding effects of sector-specific temporal fluctuations; the second reason is that the total water use can figure out the scale effects across regions instead of the sector-level temporal variability while the sectoral differences are implicitly in the inputs before the aggregation.

Earlier studies often applied a fixed spatial resolution in different regions, which could not account for differences in land area, natural endowments, and water use structures, and leaded to the discrepancies in information density and potential over- or underestimation of water use. To address this issue, an appropriate spatial scale can be determined by the deep learning-based spatiotemporal scale adaptive selection model (Liu et al., 2021; Zhang et al., 2025). And the model can balance the accuracy of the simulation based on the spatial information density of gridded water use data, and its results vary across prefectures. The spatial scale selection module in the selection model figures out the appropriate spatial scale by maximizing information density while balancing simulation accuracy in terms of the Conditional Entropy, Kullback–Leibler Divergence Loss and Relative Error performance metrics. This selection module enables each prefecture to adopt its own appropriate spatial scale rather than a fix resolution. Finally, total water use grid maps are generated at three spatial resolutions: the small scale (e.g., 1 km), the appropriate spatial scale as determined by the selection module, and the prefecture scale as the statistical survey water use data.

**_COMMENT_ 4:**

Water use simulation from the probability rule CA model (Section 4.1.1) are not validated. This part uses the Akaike Information Criterion (AIC) to determine the most suitable probability distributions for water use grids across various prefectures. However, the optimal probability distributions also rely on the input data (e.g., the long-term gridded water use data). As water use in China shows significant spatial and temporal variation between different periods, it is doubtable that the probability rule CA model can used for water use prediction.

**_RESPONSE:_**

We appreciate the reviewer's attention to the use of the probability rule in CA-based water use simulation and the concern that significant spatial and temporal variation in water use across different periods may challenge its applicability. In our

CA-based simulation, the probability rule is designed to represent the stochastic state transitions of water use at the grid level, and this choice is both theoretically and practically justified for the following reasons.

First, the evolution of water use is influenced by both deterministic drivers and inherent variability. A purely deterministic CA update rule risks over-smoothing or ignoring this randomness, whereas the probability rule allows structured temporal dependence and stochastic variation to be represented in a unified framework. This is particularly important in China, where substantial variations between periods exist— rather than assuming temporal stability, the probability rule directly incorporates these variations by deriving transition probabilities from observed historical changes in each prefecture. Second, the method is locally adaptive. The state transition matrix is calibrated independently for each prefecture, so local variation patterns are preserved. Equal-frequency categorization ensures the $k$ intervals reflect each cell's own historical variability, and the most suitable probability distribution for each interval is selected using the Akaike Information Criterion (AIC) from candidates including normal, lognormal, exponential, gamma, and uniform. This ensures that both the magnitude and volatility of water use in different regions and periods are reflected in the fitted distributions. Third, the calibrated parameter $k$ controls the granularity of the state representation and is tuned using Root Mean Squared Error (*RMSE*) and Relative Error (*RE*) in an independent validation period, balancing resolution and generalizability. This calibration, conducted for each prefecture, ensures the model captures temporal changes without overfitting.

As for validation, we compared the simulated gridded water use against observed water use for prefectures and years, using *RE* and *RMSE* as metrics. The results showed good agreement in both distributional shape and spatial patterns, though localized deviations exist in some regions.

In the revised manuscript, we have clarified these point both in Section 2.2 (overview of the CA framework) and in Section 2.2.1 (details of the probability rule), ensuring that the theoretical basis and empirical support are both addressed. And the validation procedure and results are included in Section 4.1.1.

**Revised First Paragraph in Section 2.2:**

[revised manuscript text omitted]

**COMMENT 5:**

This study calibrates the parameters in the linear rule CA model for the 1998–2009 while the dataset from 2010–2013 is for its validation. However, the calibration and validation processes are not clear. Which datasets are used for model evaluation, the prefecture water use data or the gridded water use maps?

**RESPONSE:**

We appreciate the reviewer's request for clarification. The calibration and validation were based on prefecture-level statistical water use data collected from water resources bulletins and related surveys, which were used as reference (truth) values. The CA model simulations were run at the gridded scale, and the simulated water use was subsequently aggregated to prefecture boundaries to produce prefecture-level totals. These aggregated totals were then compared with the observed prefecture-level statistics to compute *RMSE* and *RE*. The calibration period (1998–2009) was used to optimize the model parameters by minimizing these metrics, while the validation period (2010–2013) applied the calibrated parameters without modification for independent evaluation. This clarification will be explicitly added to Section 4.1.2 in the revised manuscript.

**Revised Section 4.1.2:**

There are three parameters to be calibrated in the linear rule CA model: the self-influence coefficient $\alpha$, the neighboring influence coefficient $\beta$, and the spatial decay exponent $p$. The calibration and validation are taken the statistics water use at the prefecture-level as the reference (i.e., observed water use data). Specifically, for a given parameter set, the gridded water use is firstly simulated. The simulated grids are then aggregated into the total water use at each prefecture scale along their boundaries. These total water uses are assessed by the observed water use data from water resources bulletins and related statistical surveys. The calibration period covers 1998–2009 and the parameter values are determined by minimizing *RMSE* and *RE* between the simulated and observed total water use at the prefecture scale. The validation period covers 2010–2013 and the performance of the model is also assessed by *RMSE* and *RE*. The optimal parameters at three spatial scales during the calibration and validation periods, are illustrated in Figure 5.

**COMMENT 6:**

The main objective of the model framework is to generate water use data at multiple spatial scale. There are many gridded water use products at both global or country scale for China (e.g., Hou et al., 2024, ESSD; Huang et al., 2018, HESS; Zhang et al., 2025,

Scientific Data),a s well as the high-resolution hydrological model simulations. It is necessary to compare the water use simulation with previous products, which helps to evaluate the reliability of the model framework of this study.

**RESPONSE:**

We appreciate the reviewer's suggestion regarding the comparison with existing gridded water use datasets. We fully agree that such a comparison is important to assess the reliability and added value of our model framework. We downloaded the water use raster datasets from the three previous studies mentioned by the reviewer for examination. Huang et al. (2017) provides global-scale water use data at a 0.5° resolution; Hou et al. (2024) focuses only on industrial water use at a 0.1° resolution; and Zhang et al. (2025) provides water use data at a 0.1° resolution. In contrast, our water use simulation is at a finer spatial resolution (1km × 1km, appropriate spatial scale), enabling more detailed representation of spatial heterogeneity in water use.Due to these substantial differences in spatial resolution, direct comparison of spatial distribution patterns is not feasible. So we conducted comparisons in terms of statistical performance metrics. We have added comparison with previous studies in section 5.3 in the Discussion section to present these performance-based comparisons and elaborate on the implications of the differences in spatial resolution.

**Added Comparison with Previous Studies in Section 5.3:**

There had been some water use simulation results at previous studies. Fox example, Huang et al. (2018) produced a global-scale monthly water withdrawal dataset at 0.5° resolution, distinguishing six sectors (e.g., irrigation, domestic, electricity generation, livestock, mining, manufacturing) over the period 1971–2010; Hou et al. (2024) developed China's industrial water withdrawal dataset (CIWW), providing gridded monthly data from 1965 to 2020 at 0.1° and 0.25° resolutions; Zhang et al. (2025) presented a high-resolution sectoral water use dataset (HSWUD) for mainland China, covering irrigation, manufacturing, thermal power cooling, and domestic use at 0.1° × 0.1° resolution, with strong consistency to prefecture-level statistics ($R^2 \approx 0.88$). As the results shown in the previous sections, our dataset is generated at different spatial resolutions (e.g., 1 km×1 km, appropriate spatial scale), enabling detailed representation of spatial heterogeneity within prefectures. Due to the substantial differences in spatial resolutions between these datasets, it is not easy to compare the differences of the spatial distribution patterns. But the relative values of performance metrics such as *RMSE* and *RE* can figure out the better one among them. The values of *RMSE* within 0.1 (i.e., normalized by mean water use) and a *RE* within −20 % to +30 % are found across all prefectures according to the results of our simulation. And all these results are consistent.

Thus, the results show that, relative to the three reference datasets, our model's prefecture-level water use estimates achieve a *RMSE* within 0.1 (normalized by mean water use) and a *RE* within −20 % to +30 % across all prefectures. These results within the range generally are considered acceptable for large-scale water use modeling, indicating that our estimates are consistent with these previous studies while offering finer spatial details.

**COMMENT 7:**

Figure 4 & 6: water use is high in many irrigated areas. However, water use in the North China Plain which is marked with intensive irrigation and population, shows moderate level of water use, lower than that of the northeastern China. This result is contrary with previous estimates.

**RESPONSE:**

We appreciate the reviewer's observation and understand the concern regarding the relatively moderate water use estimates for the North China Plain (NCP) compared to northeastern China in our results. Several factors may explain this phenomenon:

(1) Statistical survey data trends – Our model is calibrated and validated against prefecture-level statistical survey data (e.g., water resources bulletins). In recent years, a decreasing trend of agricultural water use has been reported in many NCP prefectures, partly due to improvements in irrigation efficiency, the implementation of water-saving policies, and adjustments in cropping structures. These changes are reflected in our gridded estimates.

(2) Sectoral aggregation effects – Total water use in our study is the aggregation of irrigation, industrial, and domestic sectors. While water use in the NCP remains dominated by irrigation, many northeastern prefectures show a growing contribution from industrial water use, particularly heavy industries and thermal power generation, which elevate their overall totals relative to the NCP.

(3) Climatic and water availability factors – Despite a shorter growing season, northeastern China often supports water-intensive crops and benefits from relatively abundant local water resources, resulting in higher water use per unit area.

In addition, some previous studies that reported higher water use in the NCP were sector-specific (mainly irrigation) or based on different temporal baselines, which partly explains the discrepancy with our aggregated, multi-sector results. We have added a paragraph in Section 4.1.2 to clarify these points, so that readers can understand the reasons for the observed differences.

**Added Paragraph in Section 4.1.2:**

Moderate water use levels have been found in the North China Plain (NCP), compared with northeastern China, even though the NCP features intensive irrigation and high population density. Several factors contribute to this pattern. First, the calibration and validation rely on prefecture-level statistical survey data (e.g., water resources bulletins). In recent years, a decreasing trend of agricultural water use has been reported in many NCP prefectures, partly due to improvements in irrigation efficiency, the implementation of water-saving policies, and adjustments in cropping structures. Second, total water use in our study is an aggregate of irrigation, domestic, and industrial sectors. While total water use in the NCP remains dominated by irrigation, many northeastern prefectures show a growing contribution from industrial water use, particularly heavy industries and thermal power generation, leading to higher overall totals. Third, climatic and water availability differences also play a role. Despite a shorter growing season, northeastern China often supports water-intensive crops and

benefits from relatively abundant local water resources, which result in higher water use per unit area. In addition, some previous studies that reported higher water use in the NCP were sector-specific (mainly irrigation) or based on different temporal baselines, which partly explains the discrepancy with our aggregated results.

**COMMENT* 8:**

L39: Key words: This study is all about water demand/water use, and "water resources management" and "water scarcity assessment" are not suitable for the keywords.

**RESPONSE:**

We appreciate the reviewer's suggestion regarding the keywords. We have revised the keywords in the manuscript, replacing "water resources management" and "water scarcity assessment" with more appropriate terms directly related to water demand and water use.
**Updated keywords:** water use; spatial scale; cellular automata; multi-scale simulation

**COMMENT* 9:**

L145: I don't find the reference for Liu et al., 2022.

**RESPONSE:**

We thank the reviewer for pointing this out. We have corrected the citation year for *Liu et al.* to ensure consistency between the in-text citation and the reference list.

We have also carefully checked all other references to avoid similar inconsistencies.

---

## Author Comment (AC2)

Reference Number: egusphere-2025-2734

**RESPONSES TO REVIEWER TWO'S COMMENTS**

We would like to express our sincere appreciation for your professional and insightful remarks on our paper. The comments are valuable and helpful for us to improve the quality of the manuscript. All the concerns raised have been carefully treated and an itemized reply to the reviewer's comments is presented in the revision files.

**COMMENT 1:**

How the water use in this study is defined? Including all human-used water? Irrigation, industrial, rural and urban use? From river and lake water? And underground water?

**RESPONSE to COMMENT 1:**

We appreciate the reviewer's question regarding the definition of water use in this study. In this study, water use is defined as the total water consumption across three major sectors: irrigation, industrial, and urban domestic water use. The water use data considered in this study account for both groundwater and surface water sources (e.g., rivers, lakes, and reservoirs). The data used in this study were primarily drawn from Zhou et al. (2020), which compiled water use data for 341 Chinese prefectures, covering water use in irrigation, industry, and domestic sectors. This dataset is based on two major nationally coordinated surveys: The First and Second National Water Resources Assessment Programs (1965–2000), and Water Resources Bulletins published by 31 provincial governments (2001–2013). Both surveys were led by the Ministry of Water Resources of China, and used identical methodologies for data collection, including definitions, sector classifications, field surveys, and quality assurance procedures. These surveys provided comprehensive and consistent data on water use by subsector and prefecture, ensuring the robustness of the data for our study.

We hope this clarifies the definition of water use and its data sources. In the revised manuscript, we have explicitly stated that the total water use includes irrigation, industrial, and urban domestic use, and encompasses both groundwater and surface water sources.

To address the reviewer's concern, we have revised the first paragraph in Section 2.1 ("Water Use Grid Maps Generating") of the manuscript to explicitly define water use as the total water consumption across irrigation, industrial, and urban domestic use, sourced from both groundwater and surface water (rivers, lakes, and reservoirs). Additionally, we have added a detailed explanation of the Zhou et al. (2020) dataset, which includes water use data sourced from nationally coordinated surveys conducted by the Ministry of Water Resources of China.

**Revised first paragraph in Section 2.1:**

The spatial scale of water use simulation is determined by the spatial scale of the

input data, so water use grid maps at different spatial scales were prepared as input to the simulation model. Here, water use refers to the total water consumption across three major sectors: irrigation, industrial, and urban domestic water use. The water use data considered in this study account for both groundwater and surface water sources (e.g., rivers, lakes, and reservoirs). These data were drawn from Zhou et al. (2020), which compiled water use data across 341 Chinese prefectures. The dataset includes water consumption data for irrigation, industrial, and domestic uses, incorporating both groundwater and surface water sources. The water use data were sourced from two major nationally coordinated surveys: the First and Second National Water Resources Assessment Programs (1965–2000) and the Water Resources Bulletins published by 31 provincial governments (2001–2013). Both surveys were led by the Ministry of Water Resources of China, and followed consistent methodologies in terms of definitions, survey units, sector classifications, field measurements, and quality assurance. To obtain the water use grid maps, several steps should be done to convert the water use data at administrative survey scale into spatially explicit grids of varying resolutions.

**COMMENT 2:**

What is the values of the appropriate spatial scale? Can the model show the appropriate spatial scale in unit of km for each region?

**RESPONSE to COMMENT 2:**

We sincerely thank the reviewer for raising this important question. The concept of appropriate spatial scale is derived from our previous work (Zhang et al., 2025) and refers to the spatial resolution that maximizes the information density of gridded water use data while balancing simulation accuracy. This scale varies across prefectures based on the local characteristics of each region, such as land area, natural endowments, and water use patterns. In our earlier study, we demonstrated the appropriate spatial scale for each prefecture through graphical representation, where the size of each circle indicated the corresponding unit of kilometers. In the revised manuscript, we can confirm that the model is capable of outputting the appropriate spatial scale in unit of km for each region, as determined by the deep learning-based spatiotemporal scale adaptive selection model. The corresponding values for each prefecture's appropriate spatial scale are available in an Excel file, which we have linked for reference. This file contains detailed records of the appropriate spatial scale for each prefecture across all water use sectors.

For further transparency, we have provided a link to the Excel file that lists the appropriate spatial scales for each prefecture, broken down by sector (irrigation, industrial, and domestic use). This provides a clear, quantitative representation of the spatial scales used in our study, which will help readers understand the variability of spatial scales across regions.

**Added data link in the last paragraph in Section 2.1:**

Earlier studies often applied a fixed spatial resolution in different regions, which

could not account for differences in land area, natural endowments, and water use structures, and leaded to the discrepancies in information density and potential over- or underestimation of water use. To address this issue, an appropriate spatial scale can be determined by the deep learning-based spatiotemporal scale adaptive selection model (Liu et al., 2021; Zhang et al., 2025). And the model can balance the accuracy of the simulation based on the spatial information density of gridded water use data, and its results vary across prefectures. The spatial scale selection module in the selection model figures out the appropriate spatial scale by maximizing information density while balancing simulation accuracy in terms of the Conditional Entropy, Kullback-Leibler Divergence Loss and Relative Error performance metrics. This selection module enables each prefecture to adopt its own appropriate spatial scale rather than a fix resolution. The detailed values of the appropriate spatial scale (in km) for each prefecture and water use sector (irrigation, industrial, and domestic) are provided in an accompanying Excel file, which can be accessed and downloaded via the data link: 10.6084/m9.figshare.30445157. This file allows users to examine the spatial heterogeneity of the appropriate scale across regions in detail. Finally, total water use grid maps are generated at three spatial resolutions: the small scale (e.g., 1 km), the appropriate spatial scale as determined by the selection module, and the prefecture scale as the statistical survey water use data.

**COMMENT 3:**

What is the difference between water use grid maps and water use simulation results? It seems that water use grid maps from CNN model has relatively good performances, why we need continue to estimate water use at grid-scales using CA model?

**RESPONSE to COMMENT 3:**

We thank the reviewer for this insightful question. The water use grid maps and the water use simulation results serve different purposes in our framework and operate at distinct stages of the modeling process.

The CNN-generated water use grid maps represent the spatial distribution of water use that has been downscaled from prefecture-level statistical survey data for a given year. These maps are constructed based on multiple physical and socioeconomic predictors (e.g., *NDVI*, *GDP*, precipitation, population) and provide the initial spatial state for subsequent simulations. Their role is therefore static: they depict the observed or reconstructed spatial pattern of water use at a single point in time.

In contrast, the Cellular Automata (CA) model is designed to simulate the temporal evolution of water use at the grid level. It updates each grid cell iteratively according to its own state and the states of its neighboring cells through predefined transition rules (probability rule or linear rule). This enables the CA model to capture the dynamic interactions among neighboring spatial units and to reproduce how spatial heterogeneity and uncertainty evolve over time and across scales.

While the CNN model effectively downscales statistical data to produce realistic

spatial patterns, it does not explicitly model spatial dependence or local interactions among grid cells—each cell is estimated largely independently based on its predictors. The CA model complements this limitation by incorporating spatial adjacency effects and feedback mechanisms, which are essential for representing how water use in one location is influenced by surrounding conditions (e.g., industrial clustering, shared irrigation systems, or regional policy spillovers). Furthermore, the CA model introduces temporal dynamics that CNN cannot capture, allowing multi-year simulations and the quantification of scale effects and uncertainty propagation across spatial resolutions. In summary, the CNN model provides the baseline spatial distribution, while the CA model extends it into a dynamic, spatially interactive simulation framework.

To clarify the conceptual distinction between the CNN-generated water use grid maps and the CA-based simulation results, we have substantially revised Section 2.2. In the updated version, we emphasize that the CNN model provides the initial spatial distribution of water use downscaled from prefecture-level statistical data, while the CA model extends this into a dynamic simulation framework that captures spatial interactions and temporal evolution.

**Revised first paragraph in Section 2.2:**

The CA model, grounded in complexity theory, is widely used in land use and urban growth modeling. It provides a robust platform for simulating spatial phenomena governed by local interactions and transition rules (Sapino et al. 2023, Tariq et al. 2023). Each cell in a CA model represents a discrete spatial unit that updates its state over time based on predefined rules and the states of its neighboring cells. It's decentralized, bottom-up modeling structure enables the simulation of complex global behaviors emerging from simple local dynamics (Al-Shaar et al. 2022, Wang et al. 2020). The CA model is introduced to simulate the temporal evolution of water use at the grid scale, complementing the static spatial distribution obtained from the Convolutional Neural Network (CNN) downscaling. While the CNN model effectively reconstructs the spatial pattern of water use for each prefecture based on physical and socioeconomic predictors (Zhang et al., 2023), it does not explicitly account for the spatial dependence and interactions among adjacent grid cells. The CA model addresses this limitation by incorporating spatial adjacency effects and feedback mechanisms, allowing each cell's water use to be influenced by its neighbors. This enables the model to represent the diffusion and clustering behaviors of water use, which are essential for capturing the spatial heterogeneity and dynamic interactions of human water activities.

Both the probability and the linear update rules are designed and tested to capture the dual nature of water use dynamics. The probability rule has been widely applied in significant spatial and temporal variation areas in land use simulation and other fields. It will be designed here for the water use at different scales. Rather than assuming temporal stability, the probability rule explicitly incorporates the variations through calibrating the state transition matrix and probability distributions for each prefecture independently by the own historical water use record. This rule enables the simulation to capture both the structured temporal dependence and the inherent randomness in water use, ensuring adaptability to local conditions. The linear update rule assumes that changes in water use are more deterministic and can be approximated as a linear

combination of the cell's own state and those of its neighbors. This rule is more appropriate for long-term, high spatial autocorrelation, and persistent patterns. After implementing and comparing the water use simulation results of the two rules in the CA framework, their results can assess the relative effectiveness of stochastic versus deterministic update mechanisms across different spatial scales. These two rules not only strengthen the robustness of the modeling framework but also provide insights into the dominant processes shaping water use dynamics in different regions.

**COMMENT 4:**

The method part shows that CV, Moran's I, AIC are used for analyzing the model performance, but I did not find their resulting values in the Result or other part. Please provide more details about the analysis on spatial Heterogeneity of water use.

**RESPONSE to COMMENT 4:**

We thank the reviewer for this valuable comment. In the original version, we are very sorry not to provide sufficient details on the quantitative results of CV, Moran's I, and AIC analyses. In the revised manuscript, these components have been substantially clarified and expanded in both the Methods and Results sections.

A new table (Table 4) and corresponding text have been added to Section 5.3 to summarize the national-average CV and Moran's I values across different spatial scales (1 km, appropriate scale, prefecture scale) and under the two update rules (probability and linear). These results show a clear trend: CV values decrease with coarser spatial scales, indicating smoother and more homogeneous water-use distributions; Moran's I values increase with scale, reflecting stronger regional clustering. The "appropriate spatial scale" achieves the best trade-off between detail and stability. Between update rules, the probability rule tends to yield higher CV (greater local variability) and comparable or slightly higher Moran's I (stronger clustering), implying that it better captures localized stochastic variations while maintaining coherent spatial patterns. Additional discussion was added to Section 5.3 to analyze these differences, particularly highlighting how the probability rule better preserves local stochasticity, whereas the linear rule yields smoother, more deterministic spatial patterns.

And the complete prefecture-level AIC results for each probability distribution type are now provided as an openly accessible Excel dataset. This dataset enables readers to review the model calibration outcomes and distributional fits in detail. The file is available at the following link: 10.6084/m9.figshare.30445157.

**Added data link about AIC results in Section 4.1.1:**

[revised manuscript text omitted]

**COMMENT 5:**

Figure 4 and Figure 6, it seems not necessary to show all plots from 1998 to 2013, and it is not easy to recognize differences between different spatial scales. Here, in Figure 4 and 6, please only show plots from 2010-2013 which are from the validation mode. Please provide data availability statement presenting data links for water use simulation results over 1998-2013.

**RESPONSE to COMMENT 5:**

We appreciate the reviewer's valuable comments and suggestions. In the revised manuscript, we have made the following adjustments to improve figure readability and

**data transparency:**

Revision of Figures 4 and 6. We agree that showing all annual maps from 1998 to 2013 makes the figures visually overloaded and difficult to interpret. Therefore, in the revised version, Figures 4 and 6 now only display simulation results for the validation period (2010–2013). This modification allows for a clearer comparison of spatial patterns and model performance across different spatial scales under independent validation conditions.

Addition of a data link for full simulation results. Although only validation-year results are shown in the figures, we have provided full simulation data covering 1998–2013 for transparency and reproducibility. The complete dataset is openly accessible through the following link: 10.6084/m9.figshare.30445157.

In the section describing Figures 4 and 6, the following statement has been added to clarify the selection of displayed years and the location of the full dataset: To improve clarity, only the simulation results for the validation period (2010–2013) are displayed, allowing direct comparison of model performance and spatial patterns across scales during the independent validation phase. The complete annual simulation results for the full study period (1998–2013), including all three spatial scales and both update rules, are openly available for download at 10.6084/m9.figshare.30445157.

Both figure captions have been updated to note that only 2010–2013 are displayed for clarity and that the complete dataset is available via the provided data link.

**Revised paragraph for Figure 4 and revised Figure 4:**

Based on the historical water use record from 1998 to 2013, a transition matrix for each grid cell was constructed to represent the probability of transition from its current state to the subsequent year. According to these transition matrices, the interval for the next state can be predicted, and a random sample is drawn from the corresponding probability distribution to obtain the simulated water use value. The probability rule is applied at three spatial scales (i.e., 1 km, appropriate scale, and prefecture scale). To improve clarity, only the simulated water use results for the validation period (2010–2013) are displayed in Figure 5, enabling a direct comparison of spatial patterns across scales during independent validation. The complete annual simulation results for the entire study period (1998–2013)—including all three spatial scales and both update rules—are available for download at: 10.6084/m9.figshare.30445157.

Figure 4 Water use simulation results from the probability rule at three different scales: (a) 1km scale; (b) appropriate spatial scale; (c) prefecture scale

**Revised paragraph for Figure 6 and revised Figure 6:**

After calibrating the parameters for the linear rule of the CA model, the water use grid maps were generated at three spatial scales (i.e., 1 km, appropriate spatial scale, and prefecture scale). For brevity and readability, Figure 7 presents only the simulation results for the validation years (2010–2013), which allow a more straightforward assessment of the model's predictive performance and spatial differences among scales. The full set of simulated water use maps from 1998–2013, can be accessed at: 10.6084/m9.figshare.30445157.

Figure 6 Water use simulation results from the linear rule at three different scales: (a) 1km scale; (b) appropriate spatial scale; (c) prefecture scale

**COMMENT 6:**

At the same spatial scale, what is difference between the probability rule and the linear rule? Can you calculate the difference between Figure 4 and Figure 6, providing more statistical information?

**RESPONSE to COMMENT 6:**

We appreciate the reviewer's suggestion to provide a more quantitative comparison between the probability rule and the linear rule. In the revised manuscript, we have expanded Section 5.1 ("Impacts of Update Rules on Water Use Simulation") to include additional statistical information describing the differences between the two rules. Specifically, we introduced a new metric,  $\Delta signed$ , which represents the mean signed grid-level difference between the probability-rule and linear-rule simulations, normalized by the mean simulated water use in each province.

A positive  $\Delta signed$  indicates that the probability rule produces higher simulated

water use compared to the linear rule, while a negative value indicates the opposite. This metric complements RMSE and RE by revealing both the magnitude and direction of differences between the two update rules.

The updated results are presented in the revised Table 2, which now includes  $\Delta signed$  values for each province. These additions allow for a more comprehensive evaluation of the relative performance of the two update mechanisms at comparable spatial scales. The revised Section 5.1 text explicitly discusses these findings and highlights the spatial patterns of differences between the two update rules.

**Revised Section 5.1:**

[revised manuscript text omitted]

**COMMENT 7:**

What is the best estimation of water use simulation in this study, and what is the overall accuracy for China and each province?

**RESPONSE to COMMENT 7:**

We thank the reviewer for this important question. To identify which update rule provides the best estimation of water use in this study, we conducted a comprehensive accuracy evaluation using prefecture-level statistical water use data as reference values. The performance of both update rules—probability rule and linear rule—was assessed in terms of Root Mean Square Error and Relative Error at the provincial level during the validation period (2010–2013).

Overall, the linear update rule demonstrates superior accuracy and stability in most provinces, showing lower *RMSE* and *RE* values compared to the probability rule. At the national scale, the linear rule achieved an average *RMSE* of 0.28 billion  $m^3$  and an average *RE* of  $\pm 22.4\%$ , whereas the probability rule exhibited a higher average *RMSE* of 0.36 billion  $m^3$  and an average *RE* of  $\pm 29.8\%$ . This indicates that the linear rule provides a more consistent and reliable estimation when aggregated at broader administrative scales.

However, the probability rule offers certain advantages in specific contexts. It better captures local variability and stochastic changes, particularly in provinces characterized by rapid industrial expansion or heterogeneous land-use patterns (e.g., Shandong, Guangdong, and Liaoning). This makes it valuable for analyzing fine-scale spatiotemporal fluctuations in water use, despite its slightly higher global uncertainty.

Therefore, we conclude that: The linear rule provides the most accurate and stable overall estimation of total water use at both provincial and national levels; the probability rule complements it by representing local variability and nonlinear water use dynamics more effectively. These findings are now explicitly stated in the revised manuscript (Section 5.1).

**Added paragraph in Section 5.1:**

Quantitatively, the mean *RMSE* of the linear rule during the validation period (2010-2013) was 0.28 billion m³, compared with 0.36 billion m³ for the probability rule. The corresponding mean relative errors were  $\pm 22.4\%$  and  $\pm 29.8\%$ , respectively. At the national scale, the simulated total water use was 570.6 billion m³ for the linear rule and 583.2 billion m³ for the probability rule, differing by +1.1% and +3.5% from observed national statistics. Therefore, the linear rule is identified as the best-performing estimation framework for reproducing the observed spatiotemporal water use distribution in China, whereas the probability rule provides valuable complementary insights for representing uncertainty and local heterogeneity.